# FineST: contrastive learning integrates histology and spatial transcriptomics for nuclei-resolved ligand-receptor analysis

Lingyu Li [1], Tianjie Wang[1], Zhuo Liang[2], Huajian Yu[1], Stephanie Ma [1,3,4], Lequan Yu [2] ✉ & Yuanhua Huang [1,2,4] ✉

Spatial transcriptomics (ST) has emerged as a powerful tool for analyzing cell-cell communication (CCC) across various biological processes, ranging from embryonic development to cancer progression. However, its limited resolution and high data sparsity hinder the detailed characterization of CCC patterns within complex tissues. Here, we introduce **FineST**, a deep contrastive learning model that leverages a histology foundation model to fuse ST and histology images, enabling **Fine**-grained **S**patial **T**ranscriptomics analysis. This approach facilitates precise nuclei segmentation, high-resolution RNA expression imputation, and the identification of intricate ligand-receptor interactions. Using both colorectal cancer VisiumHD and breast cancer Xenium datasets, we demonstrate that FineST significantly outperforms existing methods in high-resolution RNA imputation, cell type prediction, and CCC pattern discovery. With focused application to the Visium platform, FineST reveals novel biological insights into tumor-immune interactions across multiple cancer types, including invasive fronts in breast cancer, tertiary lymphoid structures in nasopharyngeal carcinoma, and PD-1 therapy resistance barriers in hepatocellular carcinoma. These findings highlight a new paradigm in ST analysis through the integration of readily available histology images.

Cell-cell communication (CCC), a fundamental form of cellular interactions, enables cells to function collectively in communities through molecular interactions, specifically ligand-receptor interactions (LRIs)[1]. Multiple computational tools have been developed to leverage the RNA levels of ligand-receptor pairs (LR pairs) to infer CCC activities, providing biological insights into diverse biological processes (e.g., maternal-fetal interface[2]) and diseases (e.g., breast cancer microenvironment[3]). However, the majority of these methods, including CellPhoneDB[4], CellChat[5] and Scriabin[6], were primarily designed for single-cell RNA-sequencing (scRNA-seq) data, omitting the physical distance between putative communicating cells. Recently,

along with the rapid development of spatial transcriptomics (ST) technologies, ST-focused methods for CCC analysis are emerging, such as SpatialDM[7], COMMOT[8] and Spacia[9], leading to more accurate analyses by considering the distance constraint of such chemical-based signaling[10]. Moreover, LIANA+ was introduced as a user-friendly framework to integrate multiple existing methods for CCC analysis[11].

Among an increasing number of ST technologies (see review[12]), sequencing-based platforms, including 10x Genomics Visium, are commonly used to study CCC for capturing a more complete picture of intercellular signaling, partly thanks to their commercial availability and high gene throughput. Despite its potential to elucidate LRIs and CCC,

[1]School of Biomedical Sciences, The University of Hong Kong, Hong Kong SAR, China. [2]School of Computing and Data Science, The University of Hong Kong, Hong Kong SAR, China. [3]State Key Laboratory of Liver Research, The University of Hong Kong, Hong Kong SAR, China. [4]InnoHK-Centre for Translational Stem Cell Biology, Hong Kong Science Park, Hong Kong SAR, China. ✉e-mail: lqyu@hku.hk; yuanhua@hku.hk

the current ST dataset is still constrained by low resolution, for example, Visium with a diameter of 55 µm covering 5 to 10 cells[13]. Moreover, the center-to-center distance between two adjacent spots in the current 10x Visium platform results in a significant portion of the tissue lacking measured gene expression[14]. While higher-resolution spatial RNA-seq techniques are becoming more accessible (e.g., VisiumHD[15] and Stereo-seq for sub-cellular resolution), the data is extremely sparse due to both biological and technical reasons, hence urgently requiring imputation or merging into a coarser bin size (e.g., 16 µm and 25 µm).

On the other hand, the histological images from the hematoxylin and eosin (HE) staining are often available from the same slide section (or consecutive slide), which provides important complementary information on the cell morphology and tissue architecture, at a super-resolution[16–18]. In digital pathology, these histology images are widely used for predicting major cell types and are also found predictive for fine-grained cell types if providing detail-annotated training data[19]. More recent research further shows its promise in predicting the expression level of individual genes[20–22], especially for those spatially relevant genes[23]. Besides this cross-sample pure-histology-based prediction, the in-sample joint analysis of histology images and ST (i.e., in a multi-modal setting) can also be highly beneficial. Particularly, the super-resolved image can be leveraged to increase the resolution of gene expression measurement in weakly supervised learning. Existing methods include XFuse based on a dual-objective variational auto-encoder[24], TESLA via graph-based smoothing[14], and iStar powered by a hierarchical image feature extractor[25].

However, three-level technical limitations hinder current histology-enhanced ST methods from broad application. First, the lack of effective image feature extractors, e.g., pre-trained Vision Transformer (ViT), makes XFuse and TESLA not able to extract the morphological features effectively. Second, the in-sample and weak-supervision setting makes it easy to overfit in such image-based imputation and lose the resemblance of the cell transcriptome profile, particularly for models that predict individual genes directly, like iStar. Third, although enhancing the resolution, none of the existing methods aims for single-cell resolution, leaving analytical challenges to users. Moreover, none of these methods focus on fine-grained LRI discovery, despite its paramount importance in ST data analysis.

To solve these challenges, in this work, we introduce FineST (Fine-grained Spatial Transcriptomics), a bimodal deep fusion framework that integrates spatial gene expression and histology image features to enhance ST analysis with fine-grained resolution and a higher signal-to-noise ratio. Critically, this method has two technical advantages: high fidelity of image-enhanced transcriptome and the nuclei-resolved LRI analytical solution. The high fidelity was achieved by using a pre-trained ViT to effectively extract morphological features and a contrastive learning scheme to map the two modalities in a low-dimensional space. The single-nuclei level resolution enhancement was enabled by the use of the histology image, consequently improving the interpretability of the detected intercellular communications. Moreover, we designed FineST as a coherent end-to-end analytic suite together with its sister tools SpatialDM and SparseAEH for rapid spatial pattern detection. Therefore, this analysis suite can accurately impute high-resolution ligand-receptor expression at a single-cell (or sub-spot) level and discover fine-grained LRIs involving multiple signaling pathways. Also, it can highlight detailed cell-cell communications for regions of interest (ROIs), especially among distinct tumor subtypes, around the tumor-immune boundary, and across differing immunotherapy response states.

## Results

### FineST model description: a contrastive learning framework enables nuclei-resolved and denoised ST with a pathology foundation model

Briefly, our bimodal model FineST contains three steps: model *training*, high-resolution *imputation*, and CCC pattern *discovery*, with

respective overviews in Supplementary Fig. 1, together with Fig. 1 and Supplementary Fig. 2. Here, we use its default data type, Visium, as an example to illustrate its two-arm inputs and model settings. For the first input arm, FineST takes spot-level histological images as square patches ([112 × 112]-pixel covering 8 × 8 tiles on an approximate [55 µm × 55 µm] spot) and employs a top-ranked pre-trained ViT, `Virchow2`[26], to extract a 1280-dim image feature/embedding vector for each segmented tile at a [14 × 14]-pixel scale (i.e., [7 µm × 7 µm], roughly a cell size) to capture fine-grained tissue characteristics (see Fig. 1A, "Methods"). For its second input aim, it takes the RNA expression count matrix of the same set of spots (see "Methods") but keeps the sub-matrix of a pre-defined gene list. By default, we exclusively include ligands and receptors for concentrating on fine-grained LRIs and CCC analysis; otherwise, we will specify the input gene list (see Supplementary Methods).

With the paired inputs, first, in the model training (Supplementary Fig. 1: ①*Training*), FineST trains four (neural network) modules jointly, particularly achieving a weakly supervised loss from image to gene expression prediction by aggregating the prediction of the 8 × 8 tiles (trans-modality loss), together with a cross-modality contrastive loss and two modality-specific reconstruction losses (Fig. 1B, "Methods"). Second, during the imputation of high-resolution gene expressions (Supplementary Fig. 1: ②*Imputation*), the trained FineST can be applied to predict the RNA expression for segmented tiles ([14 × 14]-pixel each) of both probed spots and unprobed between-spots gaps (Supplementary Fig. 2A). Besides the geometric segmentation for sub-spot resolution, we uniquely support nuclei-segmentation for single-cell resolution by using the [14 × 14]-pixel around the center of each segmented nucleus, archiving nuclei-resolved ST inference (Fig. 1A; "Methods"). Of note, to better balance information from the two modalities, the final imputation is averaged by image-based prediction and neighborhood smoothing (Supplementary Fig. 2A; "Methods"). Third, for the CCC pattern discovery (Supplementary Fig. 1: ③*Discovery*), FineST as a toolbox coherently works with our own SpatialDM[7] and SparseAEH[27] (fast reimplementation of SpatialDE[28] with sparse mathematical approximation of the large covariance matrix) to support highly scalable analyses. Therefore, six major downstream tasks can be performed at a high resolution with such enhanced signal (Supplementary Fig. 2B): 1) Identifying interacting LR pairs and local cells (for whole slide or specific regions), 2) Discovering spatial patterns shared by sub-group of the interacting LR pairs, 3) Interpreting the LR patterns via signal pathway enrichment, 4) Analyzing inter-cellular signaling from LR pair to transcription factor (TF) and target gene (TG) via specific pathways, 5) Supporting cell type prediction from reference, and 6) Examining co-localized cell types.

### VisiumHD data demonstrates FineST's superior fidelity and high scalability

To faithfully evaluate the performance of histology-imputed super-resolution ST, for the first time, we leveraged the VisiumHD platform (10x Genomics; colorectal cancer (CRC) dataset, Supplementary Tab. 1) to perform benchmarking; uniquely, VisiumHD is a technology that does not require slide registration, as it provides high-resolution ST and histology image from the same slides[15]. Here, we took the 16 µm resolution (137,051 squares) as input and aimed to impute the 8 µm resolution (545,913 squares) for achieving [2 × 2] zoom-in (Fig. 2A) and compared FineST with the state-of-the-art method iStar[25]; note, TESLA[14] and XFuse[24] were not included for technical incompatibility or inefficiency for VisiumHD data. Specifically, by comparing the observed and imputed RNA expression at 8 µm for each of the 862 ligand-receptor genes (LR genes), we found that FineST achieves reasonably good Pearson correlation coefficient (PCC; 0.507 averaged for all 862 genes), substantially outperforming iStar (0.093; Fig. 2B), as well as for structural similarity index measure (SSIM; FineST: 0.857 vs iStar: 0.824; Fig. 2C). One example is *BMP2*, a critical signaling gene[29],

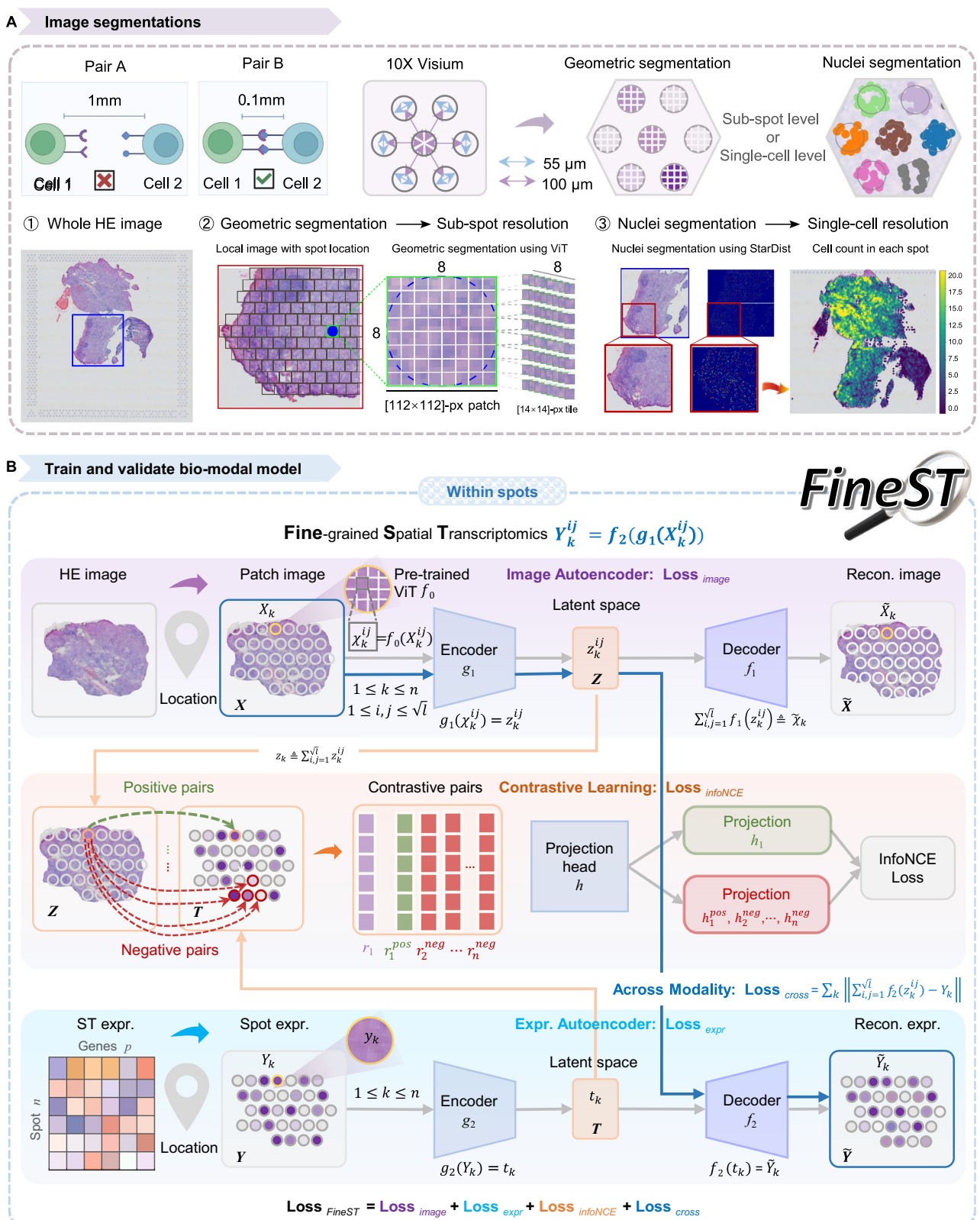

**Fig. 1 | Illustration of image segmentation and information flow in the FineST framework. A** Ligands and receptors often interact within limited spatial ranges. Here, geometric segmentation and nuclei segmentation are made in each spot to consider refined spatial distance (at the sub-spot or single-cell level) on ligand-receptor interactions. ① The whole-slide HE-stained tissue of one nasopharyngeal carcinoma (NPC) sample[38]. ② Using geometric segmentation to get sub-spot solution ST expression data (8 × 8 equal tiles, each with [14 × 14]-pixel), followed by image feature extraction with `Virchow2`[26]. ③ The nuclei segmentation results at the single-cell level, by defining [14 × 14]-pixel tiles at the center of nuclei segmentated by StarDist[53]. **B** FineST framework as an autoencoder-embedded contrastive learning model. It first performs geometric and nuclei segmentation in each spot and trains the model using paired patch images and expression data of the ligand-receptor genes (within spots; "Methods"). Part of panel (**A**) is created in BioRender. Huang, Y. (2026) https://BioRender.com/ri2o9eg.

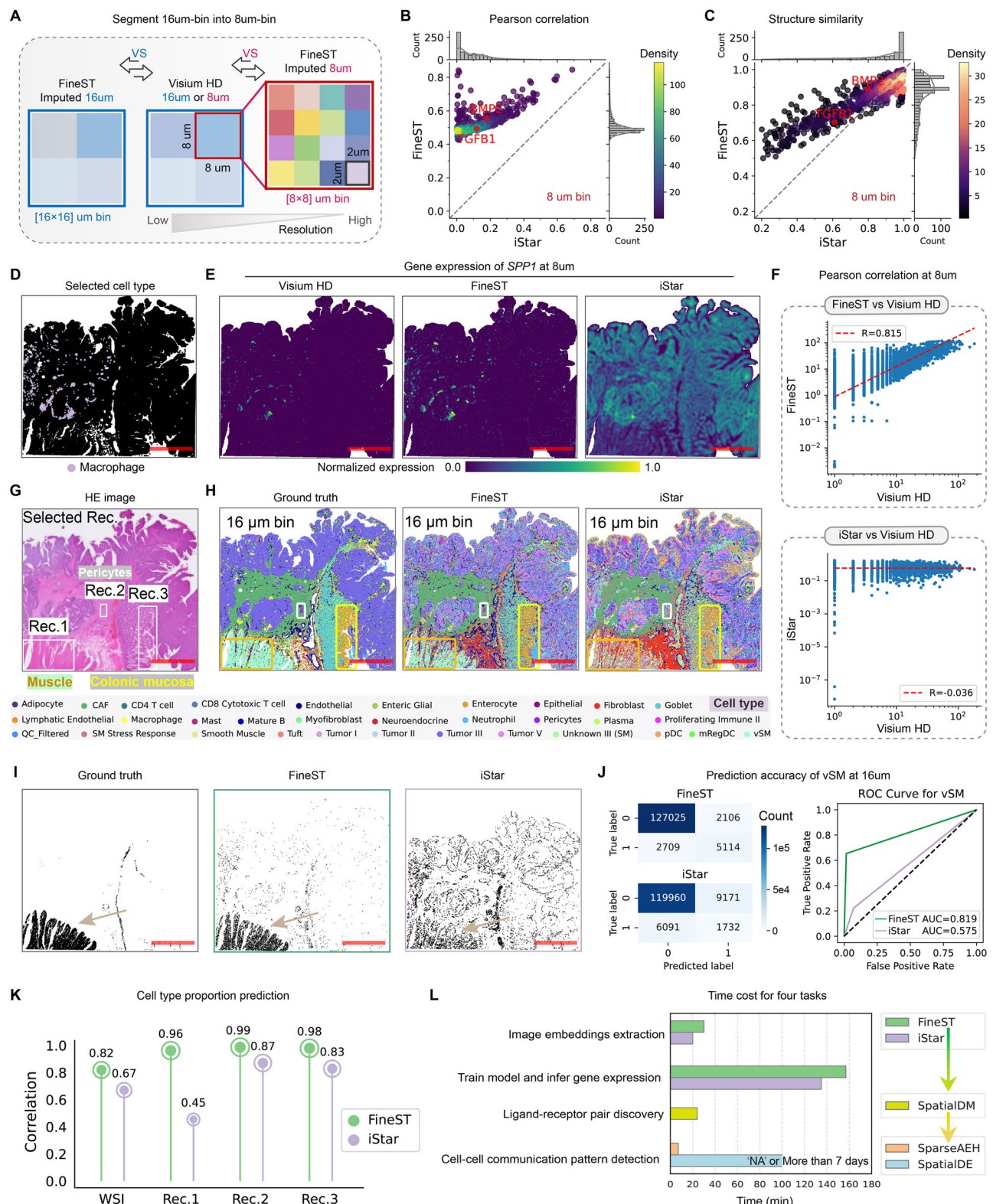

for which FineST returns a significantly higher PCC (FineST: 0.557 vs iStar: 0.177) and SSIM (FineST: 0.891 vs iStar: 0.797). This is more vividly illustrated by the *SPP1* gene, a signaling maker for macrophage, for which FineST's imputation aligns well with macrophage annotation, while iStar's imputation is overly spread (Fig. 2D–F; PCC: 0.815 vs −0.036). Such quantitative improvement is consistent when checking the reconstruction performance on this CRC dataset (16 μm; Supplementary Fig. 3A, B) or replicating iStar's demo data (a low resolution at

400 μm; Supplementary Tab. 1; Supplementary Fig. 13), indicating FineST's robust advantage in capturing the variability of LR genes, especially when the data is highly sparse. Of note, the moderate correlation level (and potentially inflated SSIM scores) on VisiumHD data may be partly caused by the more severe sparsity when increasing to 8 μm (overall zero-count: about 98%; Supplementary Tab. 2).

Next, we assessed the fidelity of the transcriptome profile by comparing its ability to use the imputed gene expressions to predict

**Fig. 2 | Benchmarking results of FineST vs iStar on a colorectal cancer dataset from VisiumHD with solid ground truth. A** Demonstration for FineST enhancing resolution from low 16 μm to high 8 μm. **B, C** Scatter plots of gene-gene Pearson correlation (**B**) and structural similarity (**C**) between FineST and iStar, for [2 × 2]-resolution enhancement of gene expression at 8 μm-binned level, where each dot represents one of 862 LR genes (diagonal line in dash represents $X = Y$). **D** Spatial distribution of macrophage cell type among the measured olorectal cancer tissue (See Supplementary Fig. 4C for more details). **E** Spatial gene expression plot of *SPP1*, a tumor-specific macrophage marker gene that is colocalized with cluster 13 from Supplementary Fig. 4C, from the ground truth (VisiumHD) and prediction value (FineST and iStar) at 8 μm resolution. **F** Pearson correlation of inferred *SPP1* expression between FineST and iStar, using 8 μm VisiumHD as ground truth. **G** HE image with three selected rectangles (abbreviated as Rec. 1, Rec. 2, and Rec. 3). One repeat is performed for the publicly available data. **H** FineST shows superior cell type prediction over iStar, especially in the three highlighted rectangles. **I** Visualization of vascular smooth muscle (vSM) cells (corresponding to Rec. 1) identified by FineST and iStar, in comparison with the ground truth[15]. **J** Confusion matrices and ROC curves for vSM cell prediction. **K** The cell type proportional correlation between the original study and imputation within the whole slide and the three color-marked rectangles (FineST vs iStar). **L** Computation time comparison in image embedding extraction (FineST vs iStar), model training and gene expression inferring (FineST vs iStar), LR pair discovery (SpatialDM), and CCC pattern detection (SparseAEH vs SpatialDE). For panels (**B**), (**C**), (**G**), (**J–L**), source data are provided in the Source Data file. Scare bars, 1 mm.

cell types. Here, we adopted the cell type annotation from the original study[15], where the 16 μm ST data were deconvolved to cell types defined in a single-cell reference atlas of this CRC dataset (Supplementary Tab. 1). Similarly, we deconvoluted cell types with the same reference atlas using FineST's or iStar's predicted 16 μm expression and compared them with the ground truth ("Methods"). Overall, both FineST and iStar capture the major cell types by large, while FineST's results better match with the original annotation, as highlighted in three detailed color-marked rectangles (Rec.1, Rec.2 and Rec.3 in Fig. 2G, H) and five key example cell types involved (Supplementary Fig. 3C), including vascular smooth muscle (vSM; Fig. 2I, J). This improvement was further evidenced quantitatively in the whole slide imaging (WSI; PCC = 0.82 by FineST vs 0.67 by iStar in Fig. 2K) and three selected ROIs from pathologist annotations[30] (ROI 1, ROI 2 and ROI 3; all with significant increase in PCC; Supplementary Fig. 3D, E), in terms of better-matched cell type proportional distribution (Supplementary Fig. 3F, G) and their cell type co-localization (Supplementary Fig. 3H) compared to the ground truth. In particular, recognizing that pathologists may focus on specific regions within HE images, FineST supports precise single-nucleus analysis in selected areas, thus accurately identifying finer cell types (Supplementary Fig. 3I). Furthermore, we also found that FineST imputed RNA expression can facilitate the cell type annotation in an unsupervised manner (Supplementary Fig. 4A, B) by strengthening the signal of key cellular gene markers (e.g., *CEACAM5*: Tumor cells, *COL1A1*: Fibroblast cells, *SPP1*: Macrophage cells; both at 16 μm and 8 μm bins), highlighting macrophages enriched in tumor boundary and cancer-associated fibroblasts (CAFs) closely surrounding tumor cells (Supplementary Fig. 4C–G). The enrichment of SPP1+ macrophages in CRC tissues correlates with tumor progression and adverse clinical prognoses. Once again, we also focus on *SPP1*, a tumor-specific macrophage marker gene reported by Oliveira et al.[15], where the signal of *SPP1* is enhanced by FineST compared to VisiumHD, but iStar suffers from ubiquitous false-positive signals. This highlights FineST's capability to accurately demarcate more aggressive tumor cells based on its proximity to SPP1+ macrophages.

Finally, taking 16 μm bin FineST-imputed data as a demonstration, our FineST-SpatialDM-SparseAEH suite can efficiently perform CCC pattern discovery (within 7 min for patch-aligned 136,954 bins, from original 137,051 bins, with a single CPU core). Such CCC pattern discovery was not possible if using the original full-covariance-based implementation (unfinished in 7 days, even with rich 40GiB memory), meaning our pipeline >1,000× speedup (Fig. 2L). FineST's robust handling of large datasets proves indispensable for deriving mechanistic insights into complex biological systems, as spatial profiling technologies advance toward generating high-resolution molecular landscapes. Leveraging FineST's effectiveness in analyzing large-scale ST data with single-cell resolution, in total, 1,253 LR pairs were detected as significant spatial co-expressed with FDR < 0.05 (Supplementary Fig. 5A, B), including *CEACAM1-CEACAM5* interaction, which activates in 16,042 bins of the whole CRC tissue (Supplementary Fig. 5C, D). By grouping these interacting LR pairs for their local interactions using SparseAEH ("Methods"), we further obtained three local patterns (Supplementary Fig. 5E), where Pattern 1 maps the CD4+ T cell region, Pattern 0 and Pattern 2 respectively enrich within the Tumor I and Tumor II cells, highlighting potential synergic functions within the tumor region. We carried out functional enrichment analysis on the LR pairs involved in different patterns. It found that WNT pathways are enriched in Pattern 0 and Pattern 2, and many of the enriched pathways are related to the biological functions in tumor cells (Supplementary Fig. 5F). Furthermore, we affirm the transmission of biological significance within these intercellular communication pathways by analyzing their downstream TFs and TGs (Supplementary Fig. 5G).

## Xenium data reveals FineST's accurate signal enhancement at nucleus resolution and differential CCC detection among three breast cancer subtypes

Besides confirming FineST's prediction accuracy at sub-spot resolution above, we further aim to demonstrate its effectiveness at single-nucleus resolution using a Xenium dataset (Supplementary Tab. 1). Xenium is an imaging-based single-molecule-resolution technology (usually) with hundreds of pre-designed RNA probes. Hence, by leveraging a paired Visium-Xenium breast cancer (BRCA) dataset from two 5 μm-adjacent sections, we simulated Xenium's spot-level gene expression according to Visium's spot size and layout to reconstruct ground-truth expression patterns[25,31] ("Methods"). Then, FineST was evaluated by predicting gene expression across an entire Visium section and benchmarking against the Xenium section, where the Xenium ground truth contains 65 shared genes across 3958 overlapping spots after registering the corresponding HE images (Fig. 3A). For computational efficiency, 957 genes (intersection of 864 LR and top 100 highly variable (HV) genes) were analyzed. FineST consistently outperformed iStar, yielding higher PCC against both Visium and Xenium ground truths (Fig. 3B; Supplementary Fig. 6A). Specifically, for 957 genes across 4992 Visium spots, mean PCC was 0.589 (vs 0.404 for iStar); for 65 overlapping genes within 3958 Xenium pseudo-spots, mean PCC was 0.616 (vs 0.542 for iStar) (Supplementary Fig. 6B, C). Notably, for the epithelial marker gene *OPRPN*, FineST more accurately recapitulated Xenium-derived ground truth than iStar, with PCC of 0.580 vs 0.050 (Visium) and 0.635 vs 0.073 (Xenium) (Fig. 3C, D), similarly for another example *EGFR* (Supplementary Fig. 6D–F).

Beyond individual gene expression inference, we further assessed FineST's accuracy across the transcriptome vector at high resolution by refining spot size from 55 μm to a near single-nucleus scale (7 μm) and predicting their cell types. Compared to Xenium-based cell type annotation, we found that FineST's pipeline achieves highly concordant cell architecture at both sub-spot and single-nucleus resolution (Fig. 3E, F). Moreover, focusing on a clinically relevant triple-positive (HER2+, ER+, PR+) ROI adjacent to adipocytes and predominantly composed of a subtype of tumor epithelium - Ductal Carcinoma In Situ 2 (DCIS 2), we demonstrated that although this region includes only five Visium spots and is classified as a mixed cluster, FineST reveals a DCIS 2-dominant composition (Fig. 3G). Remarkably, FineST, aided by nuclei segmentation, accurately

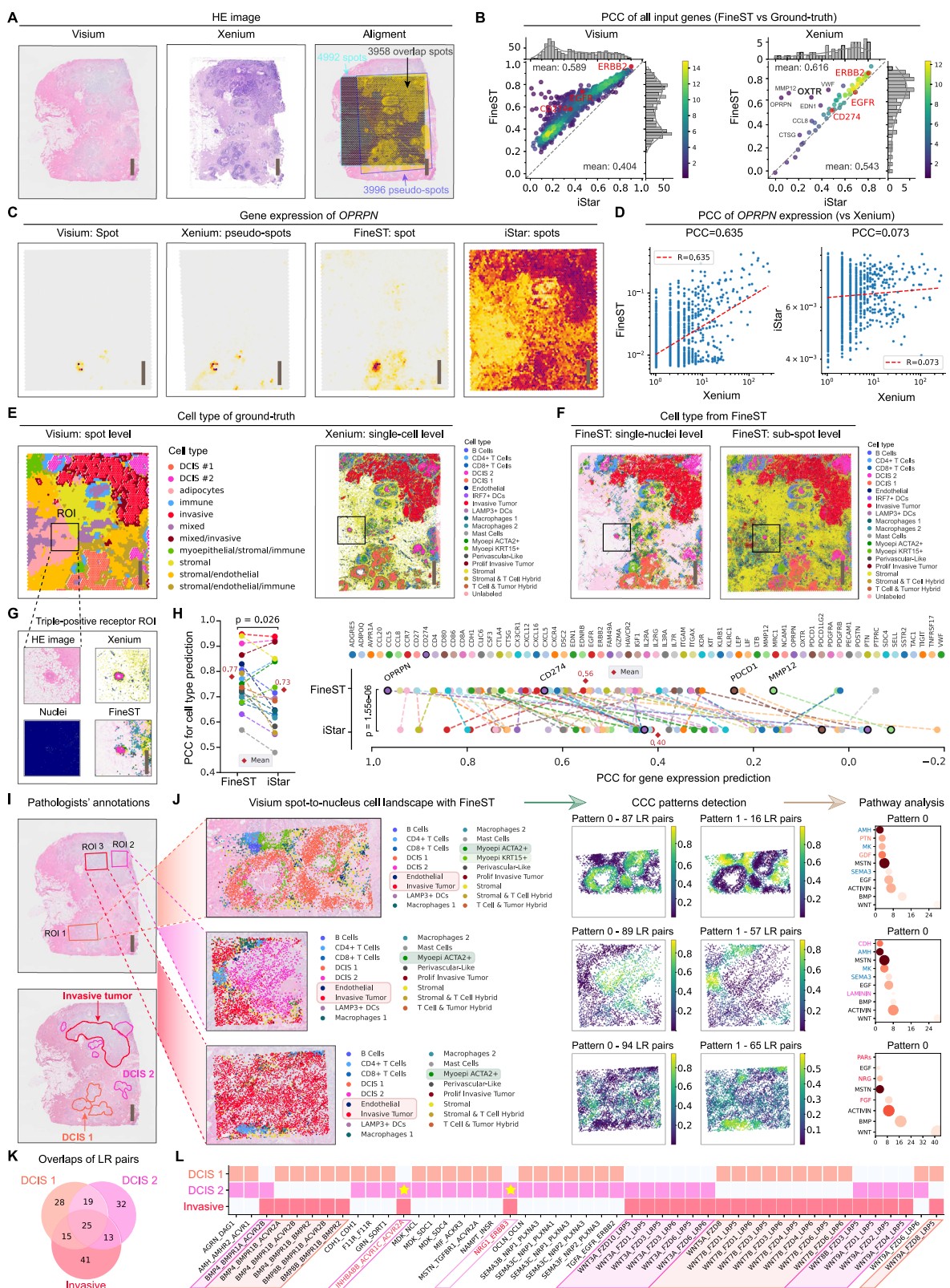

recapitulated this fine-grained cellular architecture, underscoring its ability to resolve complex tissue structure at single-nucleus resolution. To systematically benchmark performance, we input iStar results into the FineST-embedded TransImpute[32] framework for cell type prediction, and compared their aggregated 19-by-65 cell type-specific gene expression matrix to the Xenium counterpart (over 65 shared genes). As shown in Fig. 3H, FineST consistently outperformed iStar, achieving

higher mean PCCs for each gene across cell types (0.56 vs 0.40; $p = 1.55e\text{-}06$, paired two-sided $t$-test) and each cell type across genes (0.77 vs 0.73; $p = 0.026$, paired two-sided $t$-test), where gene-level examples include *CD274* (*PD-L1*, a common immunotherapeutic target) and *OPRPN* (consistent at the individual spot level, Fig. 3D).

Using FineST-enabled single-nuclei data, we further precisely predicted cell types with distinct spatial patterns linked to biological

**Fig. 3 | Nuclei-level analysis of FineST on a Visium breast cancer dataset using adjacent Xenium data as ground truth. A** HE-stained image of the Visium tissue section and adjacent Xenium section, alongside their alignment. One repeat is performed for the publicly available data. **B** Pearson correlation between FineST and iStar for all input genes, calculated after aggregating super-resolution data to spot resolution. **C** Spatial expression plots for *OPRPN* from left to right: Visium, Xenium, FineST and iStar. FineST enhances the signal relative to Visium and yields results more comparable to Xenium. **D** Pearson correlation for *OPRPN* in FineST, corresponding to panel (**C**). Each dot represents a Visium spot ($n = 4992$) or overlapping Xenium pseudo-spot ($n = 3958$). **E** Ground-truth cell type annotations at spot (Visium) and single-cell (Xenium) resolution, as reported previously[31]. **F** FineST's predicted cell types at single-nucleus and sub-spot levels. **G** FineST accurately identifies the DCIS 2 cell type in a triple-positive receptor ROI. **H** Pearson correlation for cell type abundance and mean gene expression across each cell type, comparing FineST and iStar. Each dot represents a cell type (Left, $n = 19$) or gene (Right, $n = 65$), lines connect matched pairs. Diamond indicates the mean. Statistical significance was assessed by a paired two-sided *t*-test. **I** Three marked regions (ROI 1, ROI 2 and ROI 3) dominated by DCIS 1, DCIS 2 and Invasive tumor cells. **J** Cell type deconvolution from FineST, compared with Xenium ground truth, demonstrates FineST's results visually match the ground truth and outperform Visium's lower resolution (see Supplementary Fig. 6G). Alongside, single-cell-resolved CCC patterns identified using SparseAEH (cluster number 2) and pathway enrichment analysis for Pattern 0 correspond to interesting cell distributions. **K** Venn plot of significant LR pairs (FDR < 0.05) interacting in >25% (ROI 1, 5589 cells) or >20% (ROI 2, 3330 cells; ROI 3, 5853 cells) of cells. In total, 103, 146 and 159 pairs were selected for spatial clustering analysis in the three ROIs, respectively. **L** Comparative analysis of region- and cell-specific LR pairs reveals two unique pairs specific to DCIS 2 and Invasive tumor cells. For panels (**B**), (**H**–**J**), source data are provided in the Source Data file. Scare bars, 1 mm.

significance. Figure 3I presents three ROIs from Visium HE images (DCIS 1, DCIS 2, and Invasive tumor), chosen to reflect diverse cellular compositions and demonstrate FineST's ability to dissect complex neighboring cells. Figure 3J visualizes predicted cell types, revealing detailed cellular heterogeneity. FineST's spatial cell type distributions closely match Xenium annotations, offering substantially finer granularity than Visium's spot-level clusters (Supplementary Fig. 6G). Notably, FineST accurately detects ACTA2+ myoepithelial cells in DCIS 1 and DCIS 2 with fewer false positives than spot-based methods. In DCIS 1, FineST uniquely identifies KRT15+ myoepithelial cells that were undetectable at Visium spot resolution, underscoring its superior spatial resolution. Importantly, KRT15+ myoepithelial cells appear in DCIS 1 but are nearly absent in DCIS 2 and Invasive tumor, while DCIS 2 contains more Invasive cells than DCIS 1, indicating that DCIS 2 is more invasive than DCIS 1, consistent with previously reported biological observations[31].

To investigate the underlying biological mechanisms of CCC associated with these tumor subtypes, we identified significant LR pairs within each of the above three ROIs and detected LRI patterns by setting the number of local patterns to two. As shown in Fig. 3J, Pattern 0 closely matches the distribution of our focal cell type, prompting us to analyze the pathways enriched by LR pairs in Pattern 0. We found that DCIS 1 and DCIS 2 samples are enriched in AMH, MK, and SEMA3 pathways (blue), which are absent in the Invasive. Conversely, the Invasive subtype uniquely enriches PARs, NRG, and FGF pathways, reflecting functional heterogeneity among the three tissue architectures. Beyond the shared pathways between DCIS 1 and DCIS 2, we identified ROI-specific enrichments: PTN and GDF pathways were prominent in DCIS 1, while CDH and LAMININ pathways were enriched in DCIS 2, possibly underlying their differing invasiveness potential. Comparative analysis of LR pairs across ROIs revealed 13 LR pairs shared between DCIS 2 and Invasive tumors but absent in DCIS 1 (Fig. 3K), among which two key pairs (*INHBA/INHBB-ACVR1C/ACVR2A* and *NRG1-ERBB3/HER3*) may underlie the increased invasiveness of DCIS 2 relative to DCIS 1 (Fig. 3L). Previous studies have partly supported these findings (Supplementary Tab. 3), for example, *INHBA* and *INHBB* are strongly expressed in BRCA, with high *INHBA* expression associated with the invasiveness of basal HER2+ tumor subtype and poorer survival[33,34]. In the presence of *NRG1*, *HER3* mainly heterodimerizes with *HER2*, leading to *HER3* tyrosine phosphorylation and adapter recruitment, which activates the PI3K/AKT, MAPK, and JAK/STAT pathways and promotes tumor progression[35]. While *NRG1*'s role in BRCA is established, its regulatory mechanisms remain unclear. Our results suggest that *NRG1-HER3* axis activation in invasive tumor cells may drive malignancy by triggering downstream TFs such as *SOS1* and *SOS2* (Supplementary Fig. 6H, I). Elsewhere, *ERBB3* expression correlates with increased intravasation and metastasis, with metastatic samples exhibiting higher *ERBB3* than primary tumors[36]. Here, *ERBB3* is significantly higher in DCIS 2 than DCIS 1, indicating greater

invasiveness (0.56 vs 0.39; Supplementary Fig. 6J). Notably, external validation using a BRCA Visium dataset[37] confirmed that the invasive-unique LR pair *SPP1-ITGA5-ITGB1* mainly interacts in the Invasive tumor region (Supplementary Fig. 6K). Collectively, these results demonstrate FineST's capacity to resolve intricate cellular compositions within specific ROIs, unveiling distinct tumor subtypes that are essential for understanding cancer progression and metastasis but otherwise obscured in Visium data (nor Xenium due to the lack of probing those LR genes).

## FineST identifies domain-meaningful and tumor-immune boundary ligand-receptor interactions in nucleus-resolved nasopharyngeal carcinom

Next, we showcase our FineST-SpatialDE-SparseAEH suite's full capability of nuclei-resolved CCC pattern discovery on a featured nasopharyngeal carcinoma (NPC) sample[38] (Supplementary Tab. 1). As a standard Visium dataset with poly-A selected transcriptome, the sequencing data covers the whole transcriptome and can even confirm the widespread Epstein-Barr virus (EBV) viral transcripts (Supplementary Fig. 7A, B), which the probe-based VisiumHD cannot offer. Here, in step 1, FineST enhanced the 1331 spots from 100 µm center-to-center resolution to a [7 µm × 7 µm] resolution for achieving sub-spot and single-nuclei analyses (Fig. 1A①–③). A similar sanity check of the reconstruction was performed, proving FineST's 30–70% improvement compared to iStar or TESTA (Supplementary Fig. 8A–G), including the key signaling genes *CD70* and *CD27*. Its accurate nucleus-resolved imputation can be further demonstrated by the cell type prediction (Fig. 4A, B). Quantitatively, the cell type composition confirms its better match to its scRNA-seq reference[39] than that of using its original Visium resolution (PCC: 0.64 by FineST vs 0.24 by Visium; Fig. 4C, D). FineST's unique capture of regulatory T (Treg) cells was particularly remarkable, thanks to its enhanced resolution and signal. Visually, FineST's nuclei-resolved cell type architecture offers deeper insights into the boundaries of Treg and tumor cell regions (marked in Fig. 4I), which can be further highlighted in the selected ROI in black-box (Fig. 4A', B', B", I).

Consequently, in step 2, fine-grained LRI discovery was performed (Supplementary Fig. 1: ③*Discovery*). Take the single-cell resolution as an example (similar to the sub-spot resolution), 931 LR pairs were identified as spatially co-expressed out of 1129 candidates with FDR < 0.05 (Fig. 4J, one-sided *z*-score *p*-values, Benjamini-Hochberg correction), largely more sensitive than that at the original spot resolution (332 LR pairs with FDR < 0.05). Key examples include several known NPC-related genes such as *CD70*, *CXCL16*, *MIF*, and *PVR*, with the *CD70-CD27* ligand-receptor signaling further shown for their local communication and original expression (Fig. 4K; Supplementary Fig. 7C). It has been revealed that NPC cells enhance the development and suppressive activity of Treg cells via *CD70-CD27* interaction[38]. Interestingly, a recent study has also indicated the importance of *MIF-ACKR3*

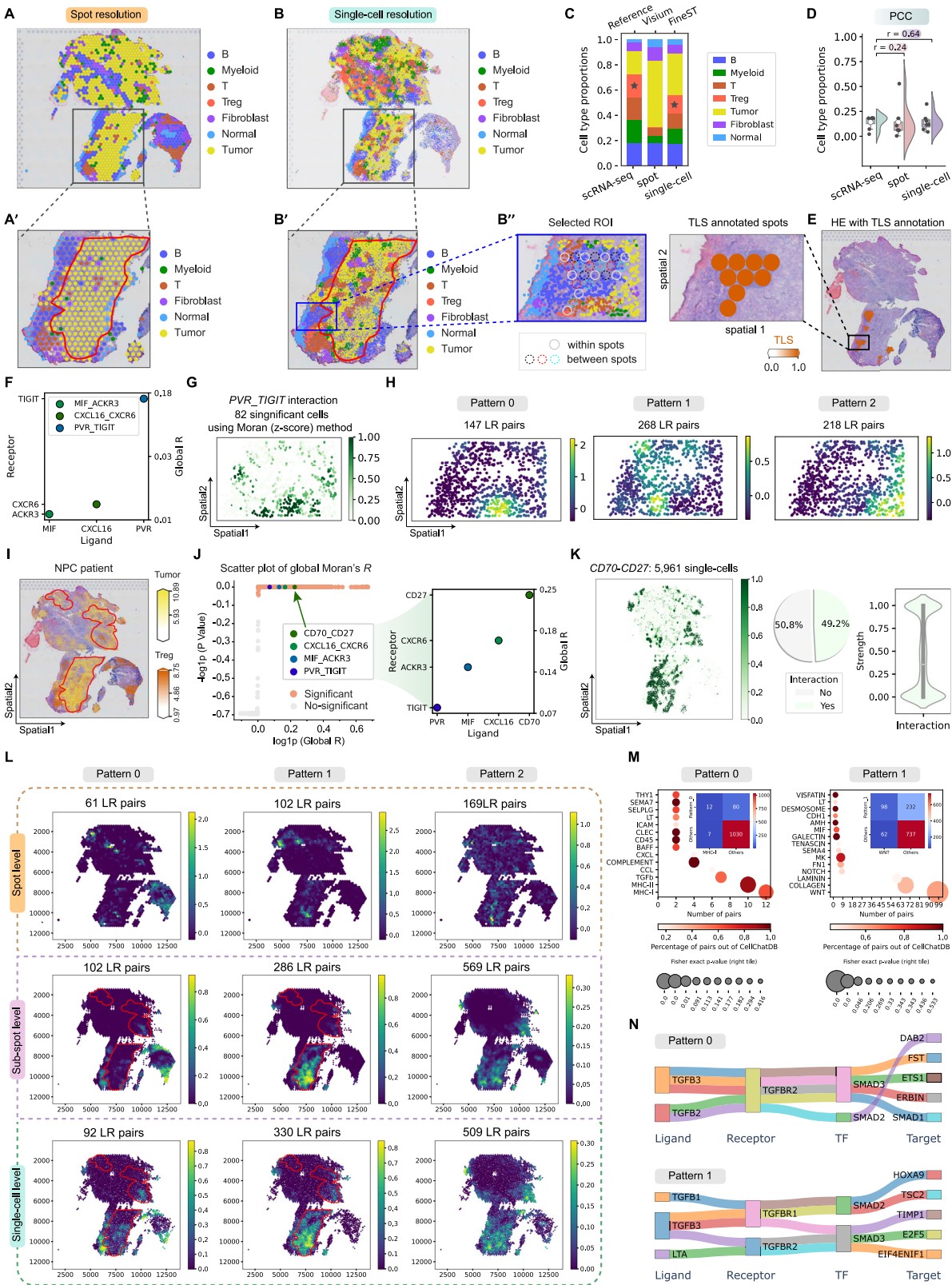

interaction for communications between B cells and fibroblast cells from a large NPC scRNA-seq cohort[40], consistent with fibroblast cells' general contributions to the tumor microenvironment by communicating with other tumor cells and by secreting extracellular matrix components[41].

In step 3, to discover patterns of intercellular communications, we further grouped the significant LR pairs by their local interaction

distributions and found three major CCC patterns (Fig. 4L; "Methods"). Impressively, the high-resolution patterns are not only more detailed but also aligned better with cell type distributions, offering critical insights into tumor-immune signaling (Fig. 4I). Specifically, the high-resolution CCC Pattern 0 corresponds to the B and T cell regions, Pattern 1 replicates the tumor cell (i.e., NPC cells) regions, and Pattern 2 moderately overlaps with the myeloid cell regions (Supplementary

**Fig. 4 | FineST applied to a nasopharyngeal carcinoma (NPC) Visium dataset reveals faithful local communication patterns at single-nucleus resolution.** **A, B** Spatial plot of seven cell types estimated by cell2location (spot resolution)[38] and FineST (single-cell resolution). **A', B'** Zoon-in views of **A** and **B**. **B"** Zoon-in view of the region marked in **B'**, where FineST increases resolution from 8 spots to 869 single cells. **C** Cell type composition from reference scRNA-seq and deconvolution at Visium spot and FineST single-cell levels. **D** Distribution and PCC of cell type proportions ($n = 7$) for Visium spot (PCC = 0.24) and FineST single-cell (PCC = 0.64) resolutions vs reference scRNA-seq. **E** Pathologists identified 36 spots (of 1331) co-localized with tertiary lymphoid structure (TLS), important for antigen presentation and T cell activation. FineST's single-cell views validate TLS by T and B cell co-localization (see **B"**). **F** Three example spatially co-expressed LR pairs detected at FineST's single-cell resolution. **G** Visualization of *PVR-TIGIT* interaction among local single cells (z-score FDR < 0.05). **H** Clustering of 633 significant LR pairs, from single-cell resolution, within selected ROI into three spatial patterns using SpatialDE. **I** Spatial co-localization of tumor and Treg cells, estimated by cell2location[38]. **J** Spatially co-expressed LR pairs detected at FineST's single-cell resolution. Scatter plot of global Moran's $R$ and one-sided z-score p-values (orange: significant, FDR < 0.05, Benjamini-Hochberg correction), with examples highlighted. **K** Communication strength of *CD70-CD27* at single-cell resolution (color: $1 - p_{localz_p}$; mean strength: 0.45, interacting cells: 5961, occupancy: 49.2%). **L** Detection of local CCC patterns by clustering significant LR pairs. Top: original spot resolution (332 pairs, SpatialDE); Middle: sub-spot resolution (957 pairs, SparseAEH) and Bottom: single-nucleus resolution (931 pairs, SparseAEH). **M** Dot plots of enriched pathways in Pattern 0 and Pattern 1 (from **L**, Bottom), 2-by-2 contingency tables for MHC-I and WNT. Statistical significance was assessed using a one-sided Fisher's exact test; dot size indicates p-value. **N** Sankey plot of selected L-R-TF-TG communication pathways. For panels (**D**, **K**), box plots show median (center), IQR (box), whiskers at 1.5 × IQR, and points for seven cell types; violin plots show density, median (white line), and IQR (thick bar). For panels (**A**), (**J**, **L**), source data are provided in the Source Data file.

Fig. 7E). Interestingly, we found the communication patterns identified by sub-spot and single-cell resolutions are largely consistent (Fig. 4L; Supplementary Fig. 7D), proving the high robustness of our model. Pathway enrichment was then performed further to interpret the identified CCC patterns. Notably, pathways related to immunity, such as MHC-I[42] and MHC-II[43], demonstrated significant enrichment in the T, B, and Treg cell regions (i.e., Pattern 0, as shown in Fig. 4M). On the other hand, the tumor cell region (i.e., Pattern 1) manifested signatures of cell proliferation[44] and NPC cell epithelial-mesenchymal transition[45], predominantly regulated by the WNT signaling pathway (refer to Fig. 4M). Also, as per the pathway enrichment results, we observed that Pattern 2 is relatively complex, where BMP, SEMA3, and ACTIVIN pathways are moderately enriched (Supplementary Fig. 7F). Additionally, the pathway enrichment, cell abundance, and L-R-TF-TG network module analyses from Visium corresponding to each pattern in Fig. 4L exhibit much lower biological interpretability compared to FineST (Supplementary Fig. 7G; Supplementary Fig. 9). Furthermore, the biological significance of these cell-cell communication pathways was demonstrated by analyzing LR pairs' downstream TFs and TGs. Figure 4N illustrates the specificity of intercellular signaling for the LR pairs within each pattern, highlighting the significant contributions of intracellular signal transmissions through receptor-transcription factor (R-TF) and transcription factor-target gene (TF-TG) pathways in promoting CCC.

Last, back to the small ROI at the peak/shift part of the boundary (Fig. 4B"), our detailed cell type annotation (distribute B, T and Treg cells) consolidates it as a tertiary lymphoid structure (TLS, Fig. 4E, annotation shared by the original authors) that is widely observed in the tumor microenvironment (TME) of various solid cancers and plays pivotal roles in shaping tumor immune phenotypes and influences the immunotherapy prognosis[46]. Furthermore, this finding was recently validated in independent primary NPC sections (P09 and P11) through colocalization analysis together with pathologists' annotation from HE staining[47] (Supplementary Fig. 10A). Within this TLS region of interest, FineST's resolution enhancement increased the number of genes captured from 9 spots to 869 single-cells (Fig. 4B", Supplementary Fig. 10B). Subsequently, we identified 633 biologically significant LR pairs out of 1129 candidates (Supplementary Fig. 10C). Similar to the entire slide results, the three LR pairs: *CXCL16-CXCR6*, *MIF-ACKR3*, and *PVR-TIGIT*, were identified as significantly co-expressed pairs again (Fig. 4F). By examining the local interaction strength, we found that the *PVR-TIGIT* interaction mainly occurred among 82 cells and was significantly enriched at the interface of tumor cells and T cells (Fig. 4G; Supplementary Fig. 10D). This finding is consistent with an earlier report about the immune inhibitory interactions between tumor cells and T cells via the *PVR-TIGIT* signaling[43]. By clustering the CCC patterns using the 633 LR pairs that interacted in more than 2 cells, we found three patterns (Fig. 4H), respectively covering about 147 LR pairs that

interacted in tumor-T/B cells (Pattern 0), followed by about 268 LR pairs interacting within tumor-B/Treg cells (Pattern 1) and about 218 LR pairs interacting in tumor-T cells and tumor-fibroblast/normal cells (Pattern 2). We also illustrated the analysis of intracellular signal transmission through the L-R-TF-TG in Pattern 2 and conducted a functional enrichment analysis for each pattern (Supplementary Fig. 10E, F). Moreover, consistent results were also found in another ROI over an extended boundary between tumor and Treg cells (Supplementary Fig. 11), double-confirming the *CXCL16-CXCR6* interaction-enabled chemoattraction between NPC tumor cells and T/B cells[43], and illustrating the *CD70-CD27*'s strong interaction along the boundary between the immune and tumor cells. These findings, made possible by our single-cell resolution, provide valuable insights into the intricate interplay among different cell types and enhance our understanding of the spatial cellular dynamics in NPC.

## FineST mitigates dropout and uncovers putative cellular cross-talk within tumor-immune barrier in hepatocellular carcinoma

Finally, we aim to demonstrate FineST's ability in characterizing the TME by directly enhancing the expression signal, particularly when a lightweight pipeline is preferred over increasing spatial resolution. Specifically, we applied it to a Visium dataset from human hepatocellular carcinoma (HCC) tissue of two patients treated with anti-PD-1 therapy, including both a responder and a non-responder (Fig. 5A, B; Supplementary Tab. 1)[48]. Here, we respectively took 1073 and 1124 genes (intersection of 911 LR and the dataset-specific top 500 HV genes) as input for the immune checkpoint blockade (ICB) non-responder (P1_T) and responder (P7_T). At the spot-level gene expression inference, FineST consistently outperformed iStar, with PCC values of 0.630 vs 0.201 and 0.567 vs 0.209 for P1_T and P7_T, respectively (Supplementary Fig. 12A, B; Supplementary Tab. 4). Take the chemokine receptor gene *CCR1* as an example, we can see that the raw Visium data suffers from signal sparsity and high dropout rates, therefore obscuring key biological signals and reducing statistical power. On the contrary, FineST effectively recovers spatial gene expression patterns of *CCR1* and enhances its correlation with SPP1+ macrophage signature score across spatial spots (Fig. 5C, D). This observation aligns with previous studies showing that *CCR1* is expressed on macrophages and promotes their migration[49], thereby providing a more robust and accurate foundation for downstream tissue architecture annotation and CCC analysis.

Leveraging FineST-enhanced data, we explored CCC mechanisms underlying immunotherapy response. Prior work has shown that interactions between SPP1+ macrophages and CAFs promote the formation of tumor immune barrier (TIB; Fig. 5A HE image), restricting immune cell infiltration in tumors[48]. Using literature-derived signature genes (Supplementary Tab. 5), our spatial analysis confirmed that SPP1+ macrophages and CAFs are co-localized specifically in ICB non-

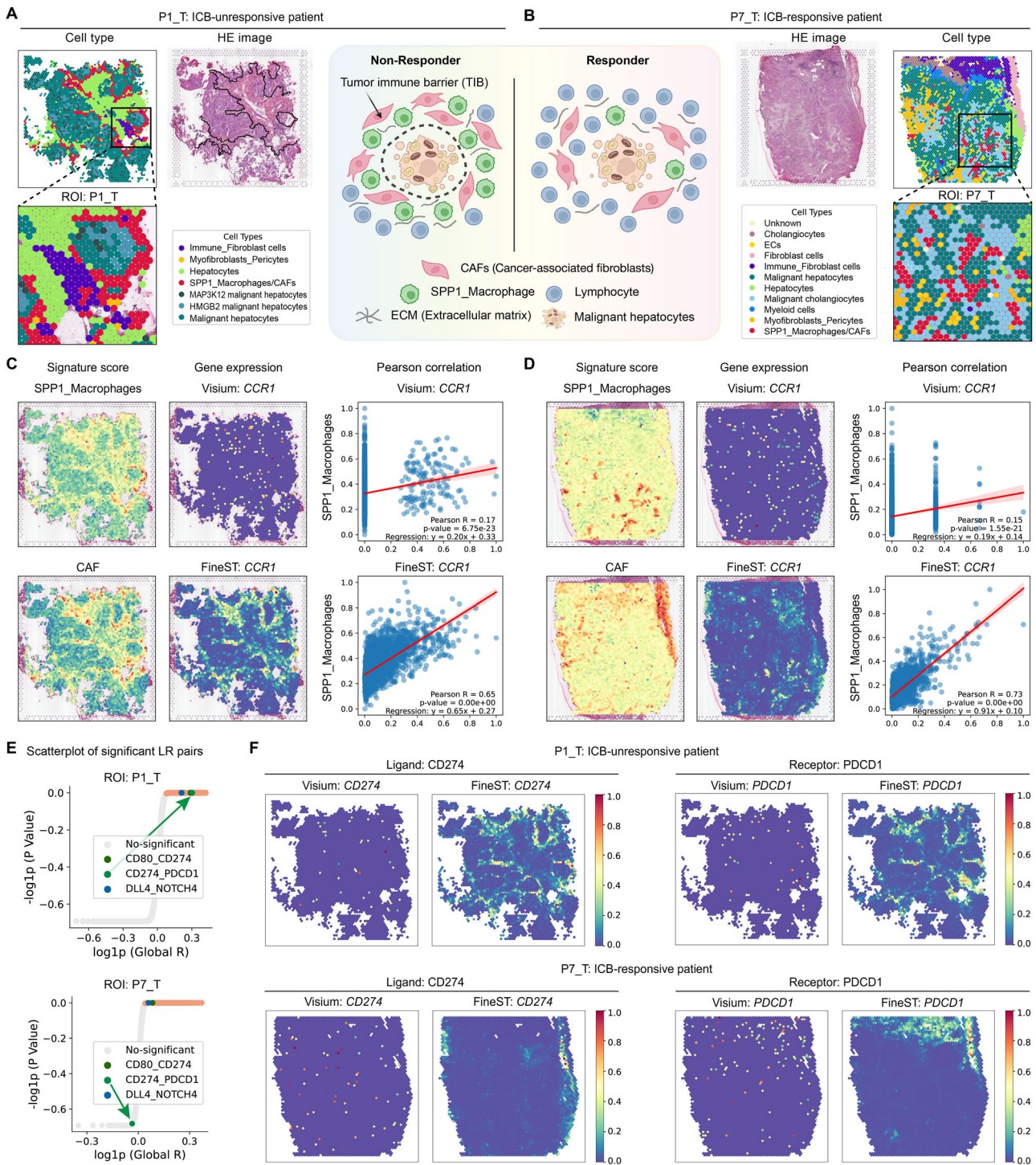

**Fig. 5 | FineST yields higher-quality, lower-noise, and biologically stronger spatial transcriptome data. A**, **B** HE staining and cell type annotation of spatial transcriptomic spots in tumor tissues from an ICB non-responder (P1_T) and responder (P7_T) from the original study[48]. In the non-responder (P1_T), a tumor immune barrier (TIB) structure is formed by SPP1+ macrophages and cancer-associated fibroblasts (CAFs). **C**, **D** Spatial signature score of SPP1+ macrophages and CAFs, spatial gene expression of *CCR1*, and Pearson correlation between SPP1+ macrophage score and *CCR1* expression across all spots (Visium vs FineST). The red line represents the fitted linear regression, and the shaded area corresponds to the 95% confidence interval. Statistical significance was assessed using a two-sided

Pearson correlation test. **E** Scatterplot of global Moran's *R* and one-sided *z*-score *p*-values for spatially co-expressed LR pairs detected using FineST-enhanced gene expression at spot-level resolution in ROI. Significant pairs (orange) were identified using FDR < 0.05 (Benjamini-Hochberg correction). The *CD274-PDCD1* interaction was detected in P1_T only. **F** Spatial gene expression of ligand *CD274* and receptor *PDCD1* across all spots (Visium vs FineST) for P1_T (non-responder) and P7_T (responder), respectively. For panels (**A**, **B**, **E**), source data are provided in the Source Data file. Part of the panels (**A**, **B**) is created in BioRender. Huang, Y. (2026) https://BioRender.com/fv0byvk.

responder (Fig. 5C). To elucidate which LR interactions between tumor and immune cells contribute to immunotherapy resistance, we applied FineST-enhanced gene expression data to identify TIB-specific LR pairs between SPP1[+] macrophages/CAFs and malignant hepatocytes (see cartoon in Fig. 5, top middle panel). Specifically, focusing on each selected ROI in P1_T and P7_T slides, we integrated relevant malignant subtypes (grouping MAP3K12 and HMGB2 malignant hepatocytes into malignant hepatocytes in P1_T, and combining malignant hepatocytes and malignant cholangiocytes in P7_T), and separately identified 555 and 583 significant LR pairs in P1_T and P7_T using SpatialDM $z$-score approach (Fig. 5E; Supplementary Tab. 7). For a sanity check, we started by examining the *CD274-PDCD1* (*PD-L1-PD-1*) interaction by which tumor cells can suppress T cells' activity and enable immune evasion[50]. Our analysis revealed that the immunosuppressive *CD274-PDCD1* interaction is active exclusively in the non-responder and with the high colocalization with TIB architecture, but absent in the responder (Fig. 5F, Supplementary Fig. 12C), consistent with the patient-level phenotype of not responding to the PD-1 inhibitor.

To further dissect the molecular basis of TIB, beyond analyzing the unique LR pairs in non-responder and responder, we further focused on the 307 overlapping LR pairs between both samples (from ROI). Of note, these LR pairs are both identified at the interface between SPP1[+] macrophages/CAFs and malignant hepatocytes, while TIB only exists in the non-responder. As an example, the *DLL4-NOTCH4* pair exhibited strong cell-cell interactions at the interface between SPP1[+] macrophages/CAFs and malignant hepatocytes in the non-responder, but this interaction was disrupted in the responder (Supplementary Fig. 12D, E). Likewise, the *WNT3-FZD9-LRP6* pair showed similar results (Supplementary Fig. 12F). These results demonstrated that these selected LR pairs significantly contribute to the immunosuppressive microenvironment of HCC, with SPP1[+] macrophages and CAFs interacting via these LR pairs to facilitate TIB structure formation.

## Discussion

In summary, we have presented FineST, a bimodal contrastive learning framework for fine-grained ligand-receptor identification, which integrates histology images and spatial RNA-seq data to enhance signal strength and achieve synchronized higher resolution. Recent advancements in imaging technologies have facilitated the integration of histology and gene expression data, playing a crucial role in understanding cellular heterogeneity and interactions within the tissue microenvironment. FineST leverages these advancements to provide a comprehensive analysis of LRIs in a spatial context. It can discover fine-grained LR pairs and detect local CCC patterns efficiently and effectively, marking a significant milestone in understanding tumor microenvironment (TME) and tumor immune barrier (TIB) structures in rational ROIs, as well as identifying the boundaries of different cell types in a spatial context. Principally, FineST can simultaneously deconvolute cell types, identify the interconnected LRI, and detect CCC for both the whole slide and specific regions. This capability is particularly relevant in the context of tissue heterogeneity, where multiple cell types and states coexist in proximity. By analyzing the spatial organization of cells and their communication networks, FineST can provide valuable insights into the functional consequences of cellular diversity and contribute to developing more effective therapeutic strategies.

In this study, we initially consider the LR genes of the whole transcriptome in this autoencoder-embedded contrastive learning model, specifically applied to the CRC and NPC datasets, as we focus on this specific task of ligand-receptor interaction detection and their spatial patterns. To further demonstrate the flexibility of our framework, we extended the analysis to include both LR genes and HV genes in the BRCA and HCC datasets. This adaptability means our model can be readily applied to any gene set of interest in ST data, such as custom gene lists and disease-associated markers. Additionally, we explored

the impact of different pre-trained ViT diversity on gene expression inference by comparing two feature extractors (`ViT256-16` and `Virchow2`), and found that `Virchow2` outperformed both `ViT256-16` and the smoothed `ViT256-16` from iStar on the NPC Visium dataset with spot resolution, achieving the highest PCC (0.323 vs 0.179 and 0.302, respectively). Briefly, the foundation model has been trained in a broad range of tissue types, and we found that they are of reasonable generalizability, particularly in the four cancer types we have tested (CRC, BRCA, NPC and HCC). Nonetheless, we suggested that users perform manual invagination before widely applying it, for example, by checking the prediction of a few contradicting markers from distinct cell types.

In the future, more efficient algorithms will be further appreciated as the size of spatial omics data is increasingly large with both higher resolution and bigger slide regions. Potential efforts could be the development of efficient mathematical approximations and lightweight but highly generalizable pre-trained models. Moreover, with the advancement of high-resolution ST technologies, the bimodal analysis will remain important, while the focus may shift from resolution enhancement to signal enhancement, which our FineST has started to show such benefits in the VisiumHD data. Importantly, FineST's contrastive learning framework is inherently flexible and can be adapted to integrate data from diverse spatial resolutions and modalities, making it suitable for new ST platforms. With minor adjustments in data preprocessing and model input formats, FineST can be extended to datasets generated by various emerging ST technologies. We anticipate that nuclei-resolved ligand-receptor analysis enabled by FineST will be increasingly valuable as ST technologies evolve, supporting finer spatial mapping and broader biological applications.

## Methods

### Datasets collection and transcriptome pre-processing

This work collected the publicly available human colorectal cancer (CRC), breast cancer (BRCA), hepatocellular carcinoma (HCC), nasopharyngeal carcinoma (NPC) and another HER2[+] BRCA (demo of iStar) datasets to benchmark FineST (Supplementary Tab. 1). First, the CRC dataset was generated with the advanced VisiumHD ST platform, which is a recent high-resolution spatial technology that utilizes a whole transcriptome probe panel and resolves data at an up to 2 μm scale within intact tissue sections[15]. Here, we conduct experiments on binned 16 μm CRC dataset to impute the 8 μm resolution-enhanced gene expression data using FineST. Taking the original binned 8 μm CRC dataset as ground truth, we can compare and evaluate FineST's performance (Fig. 2A). In total, the processed 16 μm CRC gene expression matrix contains spatial information of 137,051 squares and 18,085 genes, while the original 8 μm matrix has 545,913 squares. Second, the BRCA datasets were generated from two ST platforms: Visium (4992 spots and 18,085 genes) and Xenium (167,780 cells and 313 genes)[31]. After registration with Visium, the Xenium was aggregated into 3996 pseudo-Visium spots for ground truth analysis. Next, the HCC datasets, which include two conditions (ICB non-responder and responder), were generated with the Visium platform, with corresponding high-resolution HE images obtained from the authors[48]. Finally, the NPC dataset was generated by the 10x Genomics Visium v1 platform, converting 1331 spots and 36,601 genes[38]. Here, the ST sequencing UMI matrix and HE staining image were obtained on the same frozen section.

FineST aims to identify fine-grained LR pairs with significant spatial co-expression from a comprehensive candidate list for reliable cell-cell communication analysis. By default, we used LR lists from CellChatDB[5] (version 1.1.3) (mouse: 2022 pairs, human: 1940 pairs) as input, while users can use any customized list. Here, we focused on the LR genes, thus only keeping the 963 genes among the 1940 human LR pairs (see Supplementary Methods). For instance, after making an intersection with the LR genes and filtering out genes that are detected

in less than 10 spots or bins, 862 genes from the CRC dataset and 596 genes from the NPC dataset were used for training and prediction. The preprocessing script: `FineST.adata_LR(adata,gene_list = 'LR_genes')`. For the raw gene UMI count matrices (on the LR genes), we can perform routine preprocessing: each spot is normalized to its total count, scaled up to 10,000, and log-transformed (with adding default pseudo count 1) using Scanpy[51]. For the sake of fairness in comparison, here we feed the gene expression count matrix as input to our model, just as it is done in iStar. Users can select whether to use normalization by setting 'True' or 'False' in FineST. The preprocessing script: `FineST.adata_preprocess(adata,normalize = False)`.

**HE image segmentation for sub-spot and single-nuclei**
For benchmarking, on VisiumHD dataset with 16 μm bin, we first rescaled the HE image with the scale factor 0.5 to ensure the size of one 16 μm bin corresponds to [28 × 28]-pixel patch, thus a 8 μm bin corresponds to [14 × 14]-pixel tile (roughly about the value of 'spot_diameter_fullres' of 16 μm bin: 58 pixel and 8 μm bin: 29 pixel; Supplementary Fig. 13A). For Visium dataset, we directly segment each spot as one [112 × 112]-pixel patch, roughly about the value of 'spot_diameter_fullres' of 139-pixel (Supplementary Fig. 13B). To improve the resolution when used in gene expression inference solely from histology images, we first introduced geometric segmentation (Fig. 1A②). We will split the original spots into sub-spots as well as generate super-pixel features from the histology image. Let $X \in \mathbb{R}^H \times \mathbb{R}^W \times \mathbb{R}^3$ be the RGB-channel histology image with height $H$ and width $W$, we first partitioned the whole image into [112 × 112]-pixel image patches $X_k \in \mathbb{R}^{112} \times \mathbb{R}^{112} \times \mathbb{R}^3$ centered around each spot (Supplementary Fig. 13B: Patch). Next, each [112 × 112]-pixel image patch is further partitioned into a 8-row, 8-column grid of [14 × 14]-pixel image tiles (corresponding to [7 × 7]-μm², roughly about the size of a single cell[25]) $X_k^{ij} \in \mathbb{R}^{14} \times \mathbb{R}^{14} \times \mathbb{R}^3$ with $i, j = 1,2, \cdots, 8$ and $k = 1,2, \cdots, n$, where $n$ is the number of probed spots (i.e., the original spots, also used as "within spots").

To characterize spatial-resolved transcriptome-wide gene expression at single-cell resolution, we then introduce an alternative resolution enhancement approach, nuclei segmentation (Fig. 1A③), as a parallel framework to predict super-resolved ST data. Similar to SpatialScope[52], we also employed StarDist[53] as the default tool for nuclei segmentation. After segmentation, we denote $\hat{M}_k$ as the number of detected cells at the $k$-th spot, $k = 1,2, \cdots, m$, where $m (m \approx 4n)$ is the number of all spots (including both "within spots" and "between spots"), and then denote $\hat{M}$ as the total number of all spots calculated by $\hat{M} = \sum_{k=1}^m \hat{M}_k$. We then get a [14 × 14]-pixel image tile for each of the $\hat{M}$ cells according to the coordinators of each nucleus's location $(x_{pixel}, y_{pixel})$. Then we can have the image feature in the same format as subsopt as $X_k^{ij} \in \mathbb{R}^{14} \times \mathbb{R}^{14} \times \mathbb{R}^3$ (Supplementary Fig. 13B: Tile), with the only difference that $(i, j)$ now refers to the nucleus's location $(x_{pixel}, y_{pixel})$ instead of a tile's index. Without loss of generality, we described the model below in the sub-spot format, which can be readily applied to the single-nucleus format.

**Construction of FineST method**
FineST is a stepwise computational method to identify nucleus-resolved ligand-receptor interactions, and it consists of three components after HE image feature extraction is completed: *Training FineST on the "within spots"*, super-resolution spatial RNA-seq *imputation*, and fine-grained LR pair and CCC pattern *discovery*.

**Step0: HE image feature extraction.** We employed a pre-trained ViT $f_0$ to extract histology image features[54]. First, within each [112 × 112]-pixel image patch $X_k (k = 1, 2, \cdots, n)$, the ViT maps each [14 × 14]-pixel image tile $X_k^{ij} (i, j = 1, 2, \cdots, 8)$ into a feature vector in low-dimensional space $\mathbb{R}^C$,

$$\mathcal{X}_k^{ij} = f_0(X_k^{ij}) \in \mathbb{R}^C, \tag{1}$$

where $\mathcal{X}_k^{ij}$ represents the histology feature embedding vector of the $(i, j)$-th sub-spot within the $k$-th spot. $C$ denotes the length of the vector $\mathcal{X}_k^{ij}$, which is set to 1280 in the implementation. In our experiments, we adopted a top-ranked pre-trained model `Virchow2`[26] and skipped the fine-tuning step to improve the computation efficiency (Supplementary Tab. 6).

In case limited computing resources are available, we also provide a smaller alternative pre-trained ViT (`ViT256-16` via HIPT[54], used in iStar) for the histopathology image feature extractor (Supplementary Tab. 6). Although this approach may deliver slightly weaker inference performance than `Virchow2` (Supplementary Fig. 13), it could reduce the computational complexity with fewer splits and embedding dimensions. In contrast, for each spot, it operates on 8 × 8 tiles derived from a [128 × 128]-pixel patch (or simply, 4 × 4 tiles derived from a [64 × 64]-pixel patch) and extracts a 384-dimensional image feature vector for each segmented tile at a resolution of [16 × 16]-pixel (Supplementary Fig. 13: the star marked).

**Step1: training FineST on the within spots.** FineST will train an autoencoder-embedded contrastive learning model using the paired expression data and image feature (see Fig. 1B). As mentioned above, each observed spot $k$ will be equally divided into $l$ sub-spots ($l = 64 = [8 \times 8]$ for `Virchow2` and $l = 16 = [4 \times 4]$ for `ViT256-16` in Visium by default). Therefore, the observed data with $n$ spots contains a paired image feature matrix $\mathcal{X} \in \mathbb{R}^{nl \times C}$ and ST expression matrix $Y \in \mathbb{R}_{\geq 0}^{n \times \hat{p}}$, where we focus on $\hat{p}$ ligand and receptor genes.

**Image auto-encorder.** Firstly, one standard image embedding auto-encoder ($AE_{Image}$) is built on all patch image features of all training samples, according to cosine similarity loss function to simultaneously encode (by $g_1 : \mathbb{R}^C \to \mathbb{R}^h$) to a $h$-dim image latent space $Z$ using:

$$Z = g_1(\mathcal{X}) \in \mathbb{R}^{nl \times h}, \text{ with } Z = \{z_k : z_k \in \mathbb{R}^{l \times h}\}_{k=1}^n, z_k^{ij} = g_1(\mathcal{X}_k^{ij}) \in \mathbb{R}^h, \tag{2}$$

and reconstruct patch images (decoder by $f_1 : \mathbb{R}^h \to \mathbb{R}^C$) via learning training, i.e.,

$$\tilde{\mathcal{X}} = f_1(Z) \in \mathbb{R}^{nl \times C}, \text{ with } \tilde{\mathcal{X}} = \{\tilde{\mathcal{X}}_k : \tilde{\mathcal{X}}_k \in \mathbb{R}^{l \times C}\}_{k=1}^n, \tilde{\mathcal{X}}_k^{ij} = f_1(z_k^{ij}) \in \mathbb{R}^C. \tag{3}$$

**Transcriptome auto-encorder.** Similarly, another standard gene expression autoencoder ($AE_{Expr}$, encoder by $g_2 : \mathbb{R}_{\geq 0}^{\hat{p}} \to \mathbb{R}^h$ and decoder by $f_2 : \mathbb{R}^h \to \mathbb{R}_{\geq 0}^{\hat{p}}$) is built on all ST expression profiles of all training samples according to a Mean Squared Error (MSE) loss. It can also be expressed mathematically as follows:

$$T = g_2(Y) \in \mathbb{R}^{n \times h}, \text{ with } T = \{t_k\}_{k=1}^n, t_k = g_2(Y_k) \in \mathbb{R}^h, \tag{4}$$

$$\tilde{Y} = f_2(T) \in \mathbb{R}_{\geq 0}^{n \times \hat{p}}, \text{ with } \tilde{Y} = \{\tilde{Y}_k\}_{k=1}^n, \tilde{Y}_k = f_2(t_k) \in \mathbb{R}_{\geq 0}^{\hat{p}}. \tag{5}$$

With this procedure, the $h$-dim expression latent space $T$ can be obtained.

**Contrastive framework.** Besides directly auto-encoding the image and transcriptome modalities, we further aimed to align both modalities in their latent spaces (sub-spot-level $Z$ and spot-level $T$), where all sub-spot's embeddings within each apot are aggregated to produce a vector representation by $z_k = \sum_{i,j=1}^{\sqrt{l}} z_k^{ij} \in \mathbb{R}^h$. To do so, we introduce a contrastive learning loss to ensure high similarity between the image and expression embeddings from a paired spot (positive pair) but a low similarity if from distant spots (negative pair), as defined through the Information Noise Contrastive Estimation (InfoNCE) loss[55].

Specifically, given a batch comprising $K$ patches, we randomly select half of the patches from the training dataset. For the $k$-th patch from $K/2$ image patches, the one-to-one matched $k$-th spot from $K/2$ expression spots is considered to be a positive pair. Besides the positive patch-spot pairs in the training dataset, the remaining pairs are treated as negative pairs.

**Loss functions.** In the training step, contrastive learning employs the projections to calculate the InfoNCE loss:

$$Loss_{\text{InfoNCE}} = -\sum_{k=1}^{K} \log \frac{\exp(h_k^{\text{Image}} \cdot h_k^{\text{Expr}}/\tau)}{\sum_{l=1, l \neq k}^{K} \exp(h_k^{\text{Image}} \cdot h_l^{\text{Expr}}/\tau)}, \quad (6)$$

where $h_k^{\text{Image}} \cdot h_k^{\text{Expr}}$ ($k = 1, 2, \cdots, K$), denotes the positive pairs for the $k$-th spot, while $h_k^{\text{Image}} \cdot h_l^{\text{Expr}}$ ($l = 1, 2, \cdots, K$, $l \neq k$) denotes the negative pairs for the $k$-th spot. In this study, $K$ and $\tau$ represent the batch size and the temperature parameter, setting $K = 64$ for Visium, $K = 640$ for VisiumHD and $\tau = 0.03$, respectively.

Similarly, FineST also infers the feature matrix of image patches and the expression profile of spatial spots. On one hand, it encodes the patch image with encoder $g_1$ and then decodes the image embeddings with decoder $f_1$ through $AE_{\text{Image}}$, which is trained according to the cosine similarity loss between the rows and columns of the original and reconstrcted matrices, effectively capturing both the sub-spot-wise and gene-wise expression distributions:

$$
\begin{aligned}
Loss_{\text{Image}} &= \text{cosim}(\mathcal{X}, \tilde{\mathcal{X}}) \\
&= \frac{1}{nl}\sum_{i=1}^{nl}(1 - \cos(\mathcal{X}_{i\cdot}, f_1(g_1(\mathcal{X}))_{i\cdot})) \\
&\quad + \frac{1}{C}\sum_{j=1}^{C}\left(1 - \cos(\mathcal{X}_{\cdot j}, f_1(g_1(\mathcal{X}))_{\cdot j})\right)
\end{aligned} \quad (7)
$$

where $\mathcal{X}$ is the original patch image feature, $\tilde{\mathcal{X}}$ is the reconstructed path image feature, $\mathcal{X}_{i\cdot}$ and $\mathcal{X}_{\cdot j}$ represents the $i$-th row and $j$-th column of matrix $\mathcal{X}$, respectively (same below). On the other hand, it encodes the spot expression with encoder $g_2$ and then decodes the expression embeddings with decoder $f_2$ through $AE_{\text{Expr}}$, which is also trained according to the cosine similarity loss:

$$
\begin{aligned}
Loss_{\text{Expr}} &= \text{cosim}(Y, \tilde{Y}) \\
&= \frac{1}{n}\sum_{i=1}^{n}(1 - \cos(Y_{i\cdot}, f_2(g_2(Y))_{i\cdot})) \\
&\quad + \frac{1}{\hat{p}}\sum_{j=1}^{\hat{p}}\left(1 - \cos(Y_{\cdot j}, f_2(g_2(Y))_{\cdot j})\right),
\end{aligned} \quad (8)
$$

where $Y$ is the original ST gene expression, $\tilde{Y}$ is the reconstructed ST gene expression.

Finally, to translate from patch image to spot expression, FineST aims to encode the patch image with $AE_{\text{Image}}$ encoder $g_1$ and then decode the encodings using the $AE_{\text{Expr}}$ decoder $f_2$. Of note, the scales are different from these two modalities; therefore, we take the summation of all sub-spots after translation via $\bar{f}_2(g_1(\mathcal{X}_k)) = \sum_{i,j} f_2(g_1(\mathcal{X}_k^{i,j}))$. In this way, it can compute the following cosine similarity of cross loss function:

$$
\begin{aligned}
Loss_{\text{Cross}} &= \text{cosim}(Y, \bar{f}_2(g_1(\mathcal{X}))) \\
&= \frac{1}{n}\sum_{i=1}^{n}(1 - \cos(Y_{i\cdot}, \bar{f}_2(g_1(\mathcal{X}))_{i\cdot})) \\
&\quad + \frac{1}{\hat{p}}\sum_{j=1}^{\hat{p}}\left(1 - \cos(Y_{\cdot j}, \bar{f}_2(g_1(\mathcal{X}))_{\cdot j})\right).
\end{aligned} \quad (9)
$$

Combining Eqs. (6–9), we have the total loss function of FineST as follows:

$$Loss_{\text{FineST}} = w_1 \cdot Loss_{\text{Image}} + w_2 \cdot Loss_{\text{Expr}} + w_3 \cdot Loss_{\text{InfoNCE}} + w_4 \cdot Loss_{\text{Cross}}. \quad (10)$$

where $w_1$, $w_2$, $w_3$, and $w_4$ are weight factors employed to balance the varying loss contributions. In currently utilized datasets, all these factors were assigned a value of 1 by default. Thus, to convert a patch image feature into an ST expression profile, FineST was initially trained using the loss function represented by Eq. (10). Subsequently, the feature embedding $\mathcal{X}_k^{ij}$ of image tile $X_k^{ij}$, which belongs to the image patch $X_k$, was encoded using the encoder $g_1 : \mathbb{R}^C \to \mathbb{R}^h$. It was then decoded using the decoder $f_2 : \mathbb{R}^h \to \mathbb{R}_{\geq 0}^{\hat{p}}$. This process was performed to evaluate its performance with $Y_k$ on the test dataset, premised on the assumptions that the knowledge of the entire spot is summarized by the collective knowledge within all its sub-spots, that is, $Y_k \approx \sum_{i,j} Y_k^{ij}$ with $Y_k^{ij} = f_2(g_1(\mathcal{X}_k^{ij}))$, as shown in the information flow indicated by the blue line in Fig. 1B.

**Step2: Super-resolution spatial RNA-seq imputation.** Once the above model training finishes, we can obtain the image-based inference of gene expression for any sub-spot tile $X_k^{ij}$ via $f_2(g_1(f_0(X_k^{ij})))$. Moreover, considering the spatial neighborhood auto-correlation, we further define the imputation by balancing it with image-based inference and neighborhood smoothing. In details, for a certain sub-spot $X_k^{ij}$, let $\mathcal{N}$ be the set of top $\kappa$ nearest neighboring measured spots $Y_k$ based on the Euclidean distance metric, that is,

$$\mathcal{N}(X_k^{ij}) = \{Y_s :\| \text{Loc}(X_k^{ij}) - \text{Loc}(Y_s) \| \leq d_\kappa(\text{Loc}(X_k^{ij}))\}, \quad (11)$$

where $\text{Loc}(\cdot)$ denotes the center's 2D coordinates of any spot $k$ or sub-spot $k^{i,j}$, $\| \cdot \|$ represents the Euclidean distance ($L_2$-norm) measured on spatial sub-spot coordinates $\text{Loc}(X_k^{ij})$ and spot coordinates $\text{Loc}(Y_s)$, $d_\kappa$ means the $\kappa$-th smallest distance. Here, we set $\kappa = 6$ for Visium data and $\kappa = 4$ for VisiumHD data, while users can use any customized number. Using Eq. (11), the neighboring probed spots $Y_s(s = 1, 2 \cdots, \kappa)$ are defined, which leads to a neighborhood smoothing, as follows,

$$\sum_{Y_s \in \mathcal{N}(X_k^{ij})} w(\text{Loc}(X_k^{ij}), \text{Loc}(Y_s)) \cdot Y_s, \quad (12)$$

where $w(\text{Loc}(X_k^{ij}), \text{Loc}(Y_s))$ is the weight for a given neighboring spot expression $Y_s$, defined negatively associated with the distance from that spot to the sub-spot $X_k^{ij}$, as follows,

$$w(\text{Loc}(X_k^{ij}), \text{Loc}(Y_s)) = \frac{d(\text{Loc}(X_k^{ij}), \text{Loc}(Y_s))^{-1}}{\sum_{Y_s \in \mathcal{N}(X_k^{ij})} d(\text{Loc}(X_k^{ij}), \text{Loc}(Y_s))^{-1}}. \quad (13)$$

Finally, for each sub-spot $X_k^{ij}$, FineST gets the predicted expression value $Y_k^{ij}$ by adding the inferred expression value and the imputed expression value, i.e.,

$$Y_k^{ij} = \alpha \cdot f_2(g_1(f_0(X_k^{ij}))) + (1-\alpha) \cdot \sum_{Y_s \in \mathcal{N}(X_k^{ij})} w(\text{Loc}(X_k^{ij}), \text{Loc}(Y_s)) \cdot Y_s, \quad (14)$$

where $\alpha \in (0, 1)$ is a turning parameter used to balance the inference and imputation to achieve the best prediction, we set $\alpha = 0.5$ here (users can adjust it to a smaller value, such as 0.02, for weaker smoothness; see Supplementary Tab. 4). After that, FineST will return a super-resolution ST gene expression profile $Y = \{Y_k^{ij}\}_{k:1 \leq k \leq m}^{i,j:1 \leq i,j \leq \sqrt{l}} \in \mathbb{R}^{M \times \hat{p}}$

for each LR gene based on Eq. (14). Here, $M = m \cdot l$ is the product of the number (i.e., $m$) of all spots and the number (i.e., $l$) of grids within each spot (More details on notation representations can be seen in Supplementary Methods).

**Step3: Fine-grained LR pair and CCC pattern discovery.** In order to perform reliable CCC in high-resolution ST data and get more interpretable LRI results, we first filter out sparsely expressed ligands or receptors (only select the LR pairs that exist in more than three cells). Then, we identify LR pairs with significant spatial co-expression using the bivariate global Moran's $R$ statistics $R^{Global}$ for spatial co-expression that was supported by SpatialDM[7]. Specifically, we re-described it here as follows,

$$R^{Global} = \frac{\sum_k^M \sum_s^M w_{ks} \left(Y_k^L - \bar{X}^L\right) \cdot \left(Y_s^R - \bar{Y}^R\right)}{\sqrt{\sum_k^M \left(Y_k^L - \bar{Y}^L\right)^2} \cdot \sqrt{\sum_s^M \left(Y_s^R - \bar{Y}^R\right)^2}}, \quad (15)$$

where $Y_k^L$ and $Y_s^R$ respectively represent the normalized and log-transformed ligand and receptor expression at sub-spots $k$ and $s$. Similarly, to identify the local interacting sub-spots or single-cells for each LR pair, we also apply bivariate local Moran's $R$ statistics $R_k^{Local}$ for sub-spot or single-cell $k$ that is composed of both sender statistics and receiver statistics[7] with the definition as follows,

$$\begin{aligned} R_k^{Local} &= R_k^{Sender} + R_k^{Receiver} \\ &= \frac{Y_k^L - \bar{Y}^L}{\sigma_{Y^L} \cdot \sigma_{Y^R}} \sum_{s=1}^M w_{ks}(Y_s^R - \bar{Y}^R) + \frac{Y_k^R - \bar{Y}^R}{\sigma_{Y^L} \cdot \sigma_{Y^R}} \sum_{k=1}^M w_{ks}(Y_s^L - \bar{Y}^L), \end{aligned} \quad (16)$$

where the detailed notation representations can be found in Supplementary Methods. In fact, Eq. (16) models the interaction profiles between the $k$-th sub-spot or single-cell and its neighboring spots across all LR pairs using a two-dimensional matrix $R = \{R_k^{Local}\}_{k=1}^M \in \mathbb{N}^{M \times p}$. Based on $R$, we will use the $z$-score approach on $R_k^{Local}$ to identify significant LRI at the sub-spot or single-cell level. Lastly, we will filter the significant LRI using the adjusted $p$-value (FDR < 0.05) across the $z$-score approach. This process has been implemented in the SpatialDM package (sdm.extract_lr, sdm.spatialdm_global and sdm.sig_pairs, sdm.spatialdm_local and sdm.sig_spots functions).

Based on the local interaction hits for each LR pair, we can generate these significant LR pairs into a few distinct CCC patterns using the automatic expression histology model first proposed in SpatialDE[28], which was used in SpatialDM[7]. However, SpatialDE needs to compute and store an $M \times M$ (or $\hat{M} \times \hat{M}$) covariance matrix for high-resolution data with $M$ subpots (or $\hat{M}$ single cells), it is not well adapted for handling large datasets with tens or hundreds of thousands of samples, such as the VisiumHD dataset with 16 μm-bin or 8 μm-bin resolution. For these scenarios, within the FineST suite, we also implemented a fast version named SparseAEH that supports efficient CCC pattern discovery in analyzing high-resolution data of large sample size, reducing the computing time from days (SpatialDE) to minutes. Briefly, it leverages the principle of sparse Gaussian processes to approximate the full-covariance Gaussian likelihood with a sparse covariance matrix in a block-wise setting. Finally, we also conduct an enrichment analysis of putative pathways for each local pattern and examine the downstream transduction of LR pair signaling to interpret the local communication patterns.

**Pseudo-Visium data generation via Xenium registration**
To evaluate the super-resolution gene expression prediction accuracy, we generated spot-level pseudo-Visium data using its adjacent Xenium data (an image-based ST technique at single-cell resolution). As the available custom scripts on GitHub at https://github.com/10XGenomics/janesick_nature_comms_2023_companion described, we first registered the HE images from Visium and adjacent Xenium using a 3 × 3 transformation matrix (exported from Xenium Explorer). We then constructed a cell-by-feature matrix by interpolating spot-level bins with high-quality transcripts, assigning Xenium transcript counts for each gene to the corresponding Visium spot and coloring hexagons accordingly. Additionally, Janesick et al.[31] demonstrated a high correlation between these interpolated pseudo-Visium spots and actual Visium spots for overlapping genes.

**Evaluation criteria for super-resolution prediction performance**
To evaluate the accuracy of the predicted super-resolution gene expressions, we treated both the spot-level ground truth and the predicted gene expression at the spot level (sub-spots or single-cells were integrated back into spots). The prediction accuracy was quantified using the Pearson correlation coefficient (PCC) and the structural similarity index measure (SSIM). During computation, the ground truth matrix and the predicted gene expression matrix were decomposed into vectors for each gene. A higher PCC signifies a higher degree of similarity between the two datasets. In the SSIM calculation, for each gene, the ground truth vector $x$ and the predicted vector $x_{reconstructed}$ were flattened into a matrix based on the coordinates of spots. Subsequently, the elements of each vector were separately scaled to (0, 1), and SSIM was calculated using the function skimage.metrics.structural_similarity(x,x_reconstructed,data_range = 1). These metrics effectively capture both the global trends and the detailed spatial structures present in the super-resolution gene expression.

**Region of interesting selection and cell-type fraction computation**
In this work, we selected some ROIs based on their distinct micro-anatomical characteristics on the HE images, from the VisiumHD or Visium dataset, and then labeled them manually based on existing literature. The coordinates of ROIs were detected and aligned with ST gene expression data using the napari[56] package, which has also been plugged into SpatialData[57]. Preprocessing scripts: viewer=napari.view_image(image,channel_axis = 2,ndisplay = 2) and napari.run(). Using the coordinates of each ROI, we can identify the polygon generated by Napari (version 0.5.1) and subsequently extract the corresponding ST data. Preprocessing scripts: FineST.-crop_img_adata(roi_path,img_path,adata_path) in our FineST package. For each ROI, the overall cell-type composition across all bins or sub-spots or single-cells was then computed for each region based on the surface fractions of the locations covered by each cell type[52].

**Additional utility analyses in FineST**
**Histology clustering of significant pairs.** Here, for Visium ST data with a not very large sample size, we use SpatialDE.aeh.spatial_patterns function in SpatialDE to cluster all selected significant LR pairs into three patterns in the NPC dataset; for VisiunHD ST data with high-resolution of a large sample size, we use MixedGaussian and gaussian.run_clusters function in SparseAEH to cluster all selected significant LR pairs into three patterns in the CRC dataset. The input here is the binary matrix $X_{bin} = \{binary(1 - p_{localz_p})\} \in \mathbb{N}^{M \times M}$ of $z$-score selected sub-spots or single-cells (0 for non-significant sub-spot or single-cell, 1 for selected ones). Consistently, the calculated continuous value matrix $X_{con} = \{1 - p_{localz_p}\} \in \mathbb{N}^{M \times M}$ also can serve as input to explore interaction-level histology. Preprocessing scripts:

```
X=adata.uns['selected_spots'].astype(int)[adata.
uns['local_stat']['n_spots']>2]; gaussian=FineST.Mixed
Gaussian(adata.obsm['spatial'],group_size=16,d=30,l=
1e-5); gaussian.run_cluster(np.array(X.transpose()),3,
iter=50,init_method='k_means').
```

**Pattern consistent analysis.** FineST provided two different segmentation ways to improve the spot resolution to sub-spot resolution and single-cell resolution. We qualify the consistency between them using a Venn diagram and a heatmap. In detail, the Venn diagram is used to show the overlap of all significant LR pairs that are selected from sub-spot resolution and single-cell resolution. The heatmap is used to visualize the similarities and distinctions of pattern-based significant LR pairs selected from sub-spot resolution and single-cell, which proves the consistency of two segmentations and finally explains FineST's effectiveness and robustness in super-resolution enhancement from ST data.

**Pathway enrichment analysis.** Additionally, we also undertake pathway enrichment analysis to assess the relevance of interactions under enriched pathways in contrast to cell-type enrichment patterns. Namely, for significant LR pairs, the number of pairs associated with each pathway, as documented in CellChatDB[5] (version 1.1.3), is counted, as well as the percentage of these pairs concerning all pairs belonging to the corresponding pathway. Notably, Fisher's exact test calculates the $p$-value that the association between the queried interactions and the interactions attributed to a specific pathway is merely coincidental, which is represented by the dot size.

**Downstream transduction of LR pair signaling.** CCC is described as the transmission of intercellular signals from ligands to receptors, which are then transduced to downstream TFs via specific pathways, triggering the transcriptional response of the TGs. Exploring LR pairs, along with their downstream TFs and TGs, is crucial for understanding these intercellular communication pathways. This exploration, often neglected in previous research, involves signal amplification via R-TF and TF-TG, contributing to CCC. The DcjComm[58] database facilitates this analysis by offering a comprehensive compilation of L-R, R-TF, and TF-TG interactions based on Reactome and KEGG pathways. It incorporates human and mouse LRIs from various literature and utilizes the R-TF database, ScSeqComm[59], which includes scores indicating the degree of association between a receptor and TF within a pathway. Furthermore, here we provide an extensive repository of human and mouse TF-TG interactions using the updated RegNetwork[60]. This analysis provides a thorough view of CCC, enhancing one's understanding of cell identity and direct lineage reprogramming.

**Super-resolved spatial cell type deconvolution.** Here, all spot deconvolutions were used to classify and label spots/bins with cell types derived from the single-cell atlas. With the original ST measurement, we used the cell type deconvolution results from the original reports directly (at the original resolution). Specifically, for the NPC Visium dataset, the spatially deconvoluted cell types at the single spot level were obtained using cell2location (version 0.1) by integrating Visium spatial data with scRNA-seq reference, one integrated NPC scRNA-seq data containing 5508 cells and 33,628 genes[38]. For the CRC VisiumHD dataset, the cell types at VisiumHD 16 μm bins were identified using RCTD with the spacexr (version 2.2.1), taking a generated single-cell reference dataset containing 279,691 cells and 18,167 genes[15].

On the other hand, for our imputed gene expression, the cell type deconvolution setting is different from the original measurement for two reasons. First, our imputed values are already normalized and log transformed, making them not directly compatible with methods requiring raw UMI counts. Second, the resolution is at the cell (or sub-spot) level, making it less like a deconvolution problem and more like a classification problem. Therefore, we leveraged TramsImpute[32], a linear low-rank translation framework to perform the cell type prediction, by using the same scRNA-seq data/annotation reference as the original studies. Specifically, we used the `expDeconv` function with default parameters to run the imputed expression matrices and compared the deconvoluted cell type results with the spot-level results.

**Cell abundance and network module analysis.** To demonstrate that FineST-identified LR pairs are biologically meaningful and mechanistically distinct, we performed cell abundance and network module analyses to highlight the value of high-resolution CCC analysis. Here, cell abundance refers to the proportion of each cell type relative to the total cell abundance within each local CCC pattern. For the analysis of cell abundance within each local CCC pattern, histology clustering of significant LR pairs is performed using `gaussian.run_cluster()` with the 'mean' object, resulting in a 'cell type-by-pattern abundance' dataframe. Cell abundance within a selected pattern is then calculated using `FineST.cell_prop()`. Subsequently, a heatmap can be generated using scaled abundances across each pattern.

For the L-R-TF-TG network module of each local CCC pattern, extraction is performed using FineST and visualization is conducted in Cytoscape[61]. Specifically, after obtaining histology clustering of significant LR pairs with `gaussian.run_cluster()` using the 'labels' object, a 'LR pair-by-pattern number' dataframe is generated. For a selected pattern, the corresponding 'LRTF pathway' or 'LRTFTG pathway' is obtained using `FineST.pattern_LR2TF2TG_unique()`. This function extracts data-specific edges from RegNetwork[60], filters LR pairs enriched in the selected pattern, identifies unique ligand and receptor, traces downstream TFs of receptors using R-TF edges from ScSeqComm[59], and then screens unique TFs to collect TF-TG edges from DcjComm[58]. In this way, L-R-TF or L-R-TF-TG network modules are constructed.

**Spatial co-localization analysis.** To evaluate the spatial correlation between cell type pairs, we directly calculated the bivariate global Moran's $R$ statistic $R^{Global}$ according to Eq. (15), which has been implemented by the `Moran_R` function in SpatialDM[7] to conduct cell-cell co-localization analysis. Specifically, to investigate the efficacy of FineST in predicting spatial cell abundance of the WSI or selected ROIs, we utilized cluster maps[19] colored according to $R^{Global}$ defined in Eq. (15). The findings showcased that co-localization results are matched with the annotations from the histologist.

**Viral detection.** Epstein-Barr virus (EBV) viral transcripts detection was conducted by following a published pipeline in examining EBV in a multiple sclerosis scRNA-seq dataset[62] (github.com/huangyh09/ViralScan). Specifically, uncounted reads in the BAM file generated by SpaceRanger will be re-aligned to 833 viral genomes, including EBV, by STAR-solo v2.7.11b[63]. Then the number of UMIs will be counted for each spatial spot by using the spot barcodes. As shown in the results (Supplementary Fig. 7A, B), EBV viral transcripts were widely detected in the NPC sample (as EBV+ sample), but not seen at all in a prostate sample (as EBV− sample).

## Computational efficiency

We measured the running time for each step and presented it in Fig. 2L for the CRC data by comparing the FineST suite and the existing solutions. In general, FineST is comparable with iStar for histology-based imputation and finishes within hours. Most critically, our CCC pattern discovery (powered by our SpareAEH) is dramatically faster than existing solutions. The NPC data at the single-cell level from FineST, it contains about 40 thousand single-cells ( ≈ 40,068 single cells), where the FineST suite was over 1000 times faster than SpatialDE (SparseAEH: ≈ 7 min vs SpatialDE: > 7 days). For the end-to-end analysis of a typical VisiumHD dataset ( ≈ 100 thousand bins), FineST usually finishes within 30 mins, while SpatialDE does not work (job killed for huge memory consumption). Experiments were conducted on an NVIDIA A100-PCIE-40GB graphics card.

## Reporting summary

Further information on research design is available in the Nature Portfolio Reporting Summary linked to this article.

## Data availability

The VisiumHD data and Chromium Single Cell Gene Expression Flex of human colorectal cancer (CRC) were downloaded from the 10x Genomics datasets here: https://www.10xgenomics.com/products/visium-hd-spatial-gene-expression/dataset-human-crc with 'Visium HD, Sample P2 CRC' and 'Chromium Single Cell Flex, aggregated' Files. The raw and processed Visium, Xenium spatial sequencing data, and Chromium Single Cell Gene Expression Flex of human breast cancer (BRCA) tissues were downloaded from 10x Genomics https://www.10xgenomics.com/products/xenium-in-situ/preview-dataset-human-breast with 'Visium Spatial', 'In Situ Sample 1, Replicate 1' and 'FRP' Files, and GEO under accession number GSE243280 with 'GSM7782699', 'GSM7780153' and 'GSM7782698' Samples. Cell type annotations for Xenium, Visium, and Chromium Flex datasets of BRCA are available for download in the 'Cell Type Annotations' Section. The raw and processed Visium spatial sequencing data of human hepatocellular carcinoma (HCC) tissues were downloaded from Mendeley Data under accession number skrx2fz79n [https://data.mendeley.com/datasets/skrx2fz79n/1], while the cell type annotations and high-resolution HE-stained images were provided by the original corresponding author. The raw and processed Visium spatial sequencing data of human primary nasopharyngeal carcinoma (NPC) tissues were downloaded from GEO under accession number GSE200310. The integrated scRNA-seq data of NPC were obtained from our colleagues and are available from the corresponding author upon request. For easier reuse, we also included them in the FineST Python package as follows: the CRC data: `FineST.datasets.CRC16um()`, `FineST.datasets.CRC08um()`, the BRCA data `FineST.datasets.BRCA()`, the HCC data `FineST.datasets.HCCP1T()`, `FineST.datasets.HCCP7T()`, and the NPC data `FineST.datasets.NPC()`. The ligand-receptor databases are available from the CellChat repository: https://github.com/sqjin/CellChat/tree/master/data. Source data are provided with this paper.

## Code availability

The FineST algorithm is a publicly available Python package at https://github.com/StatBiomed/FineST and https://doi.org/10.5281/zenodo.8343616[64]. Detailed documentation and analysis notebooks to reproduce results in this paper are also included in this repository (https://finest-rtd-tutorial.readthedocs.io). All data analyzed in this work are available through the figshare link: https://figshare.com/articles/dataset/FineST_supplementary_data/26763241.

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

## Acknowledgments

We kindly thank Dr. Lanqi Gong and Dr. Wei Dai from the Department of Clinical Oncology, the University of Hong Kong, for sharing the NPC dataset and providing biological insights. We also thank Dr. Youqiong Ye and Dr. Zhenzhen Xun from the School of Medicine, Shanghai Jiao Tong University, for sharing high-resolution HE-stained images and cell type annotations of the HCC dataset. We are grateful to StatBiomed lab members, especially Dr. Shumin Li for the contrastive model and Dr. Ruiyan Hou for package efficiency. This project is supported by the Research Grants Council of the Hong Kong SAR, China (grant numbers T12-705-24-R, YCRG-C7004-22Y, and 17126725), the National Natural Science Foundation of China (grant number 62222217), the InnoHK initiative of the Innovation and Technology Commission of the Hong Kong Special Administrative Region Government, and the University of Hong Kong through a startup fund and a seed fund (Y.H.), a PDF scheme (L.L.) and a Postgraduate Scholarship (T.W.). This project is also supported in part by the Research Grants Council of Hong Kong (17200125 and T45-401/22-N) (L.Y.), the Research Grants Council of Hong Kong (Research Fellow Scheme RFS2122-7S05) and Croucher Foundation Senior Research Fellowship (S.M). S.M. is also a Jimmy and Emily Tang Professor in Molecular Genetics.

## Author contributions

Y.H. conceived and supervised this study with support from L.Y. L.L. implemented the FineST and performed all data analysis. T.W.

implemented SparseAEH for rapid CCC pattern identification. Z.L. and L.Y. assisted with the image modeling. H.Y. and S.M. provided valuable biological insights on HCC. L.L. and Y.H. wrote the manuscript.

## Competing interests

The authors declare no competing interests.
