## [Transparent Peer Review file · Nature Communications]

FineST: Contrastive learning integrates histology and spatial transcriptomics for nuclei-resolved ligand-receptor analysis

Corresponding Author: Professor Yuanhua Huang

Version 0:

Reviewer comments:

Reviewer #1

(Remarks to the Author)

The revision has addressed my concerns.

(Remarks on code availability)

Reviewer #2

(Remarks to the Author)

The authors have worked hard on improving the manuscript describing and benchmarking FineST.

I still think that a focus on more modern ST methods would increase the impact of this work and the method's range, but the authors are free to keep their focus on Visium1 if they choose to.

I don't have any further remarks for this manuscript.

(Remarks on code availability)

Point-by-point response

The responses are highlighted in blue.

Reviewer #1 (Remarks to the Author):

The revision has addressed my concerns.

Response: We thank the reviewer for the positive evaluation and for recognizing our revisions and efforts.

Reviewer #2 (Remarks to the Author):

The authors have worked hard on improving the manuscript describing and benchmarking FineST.

I still think that a focus on more modern ST methods would increase the impact of this work and the method's range, but the authors are free to keep their focus on Visium1 if they choose to.

I don't have any further remarks for this manuscript.

Response: We thank the reviewer for the constructive assessment of our work and for acknowledging the improvements made to the manuscript. In response to the reviewer's suggestion regarding broader applicability to newer spatial transcriptomics (ST) platforms, we have added a brief discussion (outlining how FineST could be adapted to new ST platforms) in the Discussion as follows:

“Importantly, FineST’s contrastive learning framework is inherently flexible and can be adapted to integrate data from diverse spatial resolutions and modalities, making it suitable for new ST platforms. With minor adjustments in data preprocessing and model input formats, FineST can be extended to datasets generated by various emerging ST technologies. We anticipate that nuclei-resolved ligand-receptor analysis enabled by FineST will be increasingly valuable as ST technologies evolve, supporting finer spatial mapping and broader biological applications.”